# A Review of Event Deposits in Lake Sediments

**Pierre Sabatier** [1,*] , **Jasper Moernaut** [2] , **Sebastien Bertrand** [3] , **Maarten Van Daele** [3] , **Katrina Kremer** [4,5] , **Eric Chaumillon** [6] and **Fabien Arnaud** [1]

1   EDYUTEM, CNRS, Université Savoie Mont Blanc, Bâtiment Pôle Montagne, 5 bd de la mer Caspienne, 73376 Le Bourget du Lac, France; fabien.arnaud@univ-smb.fr
2   Department of Geology, University of Innsbruck, Innrain 52f, 6020 Innsbruck, Austria; jasper.moernaut@uibk.ac.at
3   Renard Centre of Marine Geology, Ghent University, Krijgslaan 281, S8, 9000 Gent, Belgium; sebastien.bertrand@ugent.be (S.B.); maarten.vandaele@ugent.be (M.V.D.)
4   Swiss Seismological Service, ETH Zurich, Sonneggstr. 5, 8092 Zurich, Switzerland; katrina.kremer@sed.ethz.ch
5   Institute of Geological Sciences & Oeschger Centre for Climate Change Research, University of Bern, Baltzerstrasse 1+3, 3012 Bern, Switzerland
6   LIENSs CNRS, Université de la Rochelle, Bâtiment Marie Curie, Avenue Michel Crépeau, 17042 La Rochelle, France; echaumil@univ-lr.fr
*   Correspondence: pierre.sabatier@univ-smb.fr

**Abstract:** Event deposits in lake sediments provide invaluable chronicles of geodynamic and climatic natural hazards on multi-millennial timescales. Sediment archives are particularly useful for reconstructing high-impact, low-frequency events, which are rarely observed in instrumental or historical data. However, attributing a trigger mechanism to event deposits observed in lake sediments can be particularly challenging as different types of events can produce deposits with very similar lithological characteristics, such as turbidites. In this review paper, we summarize the state of the art on event deposits in paleolimnology. We start by describing the sedimentary facies typical of floods, glacial lake outburst floods, avalanches, hurricanes, earthquakes, tsunamis, volcanic eruptions, and spontaneous delta collapses. We then describe the most indicative methods that can be applied at the scale of lake basins (geophysical survey, multiple coring) and on sediment cores (sedimentology, inorganic and organic geochemistry, biotic approach). Finally, we provide recommendations on how to obtain accurate chronologies on sediment cores containing event deposits, and ultimately date the events. Accurately identifying and dating event deposits has the potential to improve hazard assessments, particularly in terms of the return periods, recurrence patterns, and maximum magnitudes, which is one of the main geological challenges for sustainable worldwide development.

**Keywords:** event deposits; lake sediments; trigger processes; turbidites; homogenite; proxies

## 1. Introduction

Natural hazards pose serious threats to human societies. Most events can be grouped into two main categories: geodynamic (e.g., earthquakes, tsunamis, volcanic eruptions) and climatic (e.g., floods, hurricanes, avalanches). Robust knowledge about return periods, recurrence patterns, and future trends of such natural hazards in a given location is thus one of the main scientific challenges for sustainable worldwide development [1,2]. However, the spatial and temporal distribution of extreme climate events may be rather heterogeneous [3,4], and it is thus difficult to generalize from an array of observation sites. Additionally, the effects of ongoing climate change on the frequency and intensity of extreme events and the lack of long-term instrumental data preclude the identification of significant trends for the rarest climate events [5,6]. Indeed, even if historical accounts can provide interesting data for hazard assessment, in most regions, extreme events are too rare and/or historical data are too sparse and of a limited time span, or cover too short periods,

hence, precluding the identification of return periods for the most disastrous events [7,8]. As naturally widespread geological recorders, lake sedimentary basins, from high-altitude environments to coastal regions, provide invaluable archives that potentially permit overcoming this issue by providing plurimillennial-long archives (Figure 1) of a large variety of events, such as floods (e.g., [9]), glacial-lake outburst floods (GLOFs; e.g., [10]), hurricanes (e.g., [11]), avalanches (e.g., [12]), earthquakes (e.g., [13]), tsunamis (e.g., [14]), or volcanic eruptions (e.g., [15]).

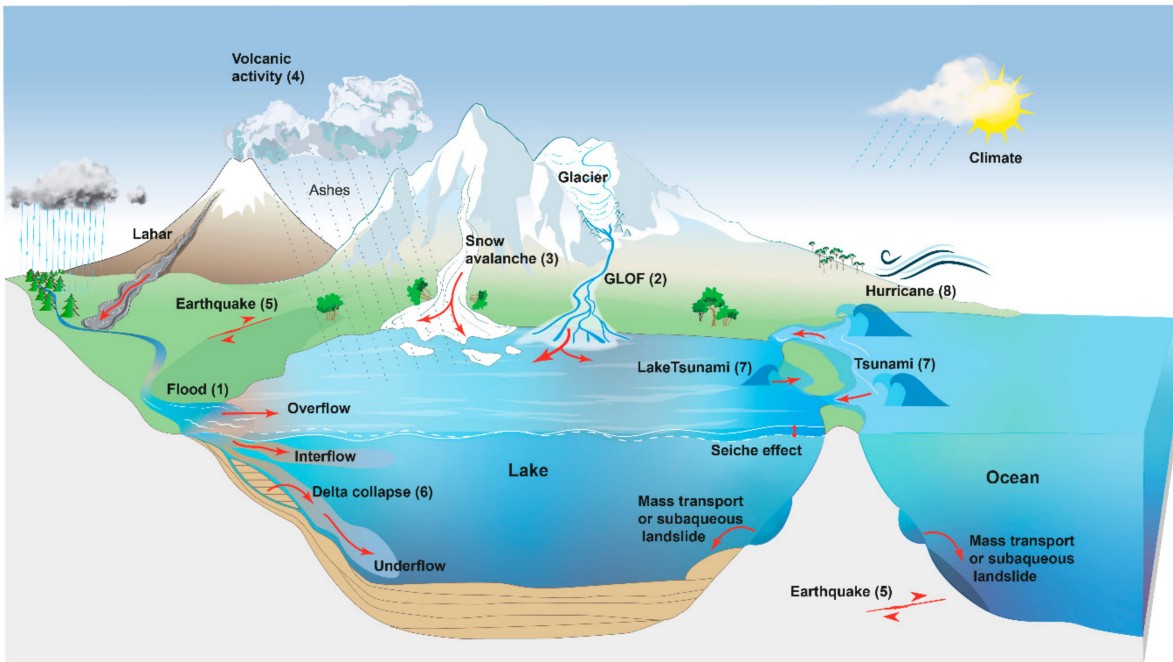

**Figure 1.** Processes at the origin of event deposits in lake systems. The following groups of events are illustrated: (1) floods, (2) glacial-lake outburst floods (GLOFs), (3) snow avalanches, (4) volcanic eruptions, (5) earthquakes, (6) delta collapses, (7) marine or lacustrine tsunamis, and (8) hurricanes.

The first step in reconstructing event deposit chronicles from lake sediments is to identify and characterize event deposits, followed by the attribution of a sedimentary facies and/or deposit geometry to a triggering mechanism. Indeed, a given lake sediment sequence often records a combination of natural hazards (Figure 1), such as floods and earthquakes, e.g., [16–20], floods and avalanches [12], floods and GLOFs [21], floods and tsunamis [22], hurricanes and tsunamis [16], earthquakes, volcanic eruptions, and floods [23], etc. These event deposits are generally recorded as turbidites, debrites, homogenites, slumps, and mass-transport deposits, which each have their own sedimentological signature. Hence, many of these event deposits resulting from natural hazards are related to gravity-driven flows. Since the reference paper by Bouma (1962) [24] provided the first model to explain the relationship between flow conditions, sedimentary structures, grain-size variations, and associated depositional processes of turbidites, new facies models for coarser and finer-grained density-flow deposits have been compiled [25,26], and more recent papers have provided a precise terminology associating flow processes, sediment concentration, and the composition of the current deposit [27,28]. Such depositional models thus allow the extraction of information about the flow conditions based on the deposits but deciphering the initial trigger of these gravity flows can be particularly challenging. For example, turbidity currents that were initiated by different triggers can often produce very similar lithological expressions, rendering the identification of the trigger often difficult [17,18,29,30].

During the last two decades, an increasing number of studies have succeeded in relating lacustrine event deposits to their triggers using a variety of techniques. In this

review paper, we summarize the state of the art concerning event-deposit identification, characterization, and attribution to specific triggers. We first define the most important geological processes that may produce event deposits in lake systems, leading to specific sedimentary facies. Then, we describe different types of indicative proxies, such as spatial imprint, lithofacies, and biological and organic facies, which can be used to precisely characterize event deposits and determine their triggering mechanism. Finally, we outline good practices in terms of chronology for the study of sedimentary sequences containing event deposits and give some recommendations.

## 2. Processes and Associated Sequences

### 2.1. Flood

A flood event usually implies overland runoff and larger-than-average river discharge, which induce strong erosion and, in turn, particle transfer from the source area to natural sinks, such as lakes. Floods are among the most frequent and devastating natural disasters worldwide [31]. In the context of climate change, the rapid and hardly predictable global reorganization of air-mass circulation disturbs the usual distribution of precipitation patterns that generate floods, making them some of the least predictable impacts of climate change [32]. It is thus crucial to understand the relationship between past climate changes and flood regimes. In that sense, lake sediments are generally considered as one of the best natural archives to reconstruct flood chronicles beyond the historical period [33].

As areas of null slope along a river or stream, lakes trap the large majority of the river-borne sediment that enters them. The in-lake distribution of the incoming sediment-laden river water depends on its relative density compared to the lake water. River-water density is mostly related to its temperature and sediment load. In an iconic paper linking this behavior with sediment facies, Strum and Matter [34] proposed three cases (Figure 2):

1. If the incoming current is less dense than the lake surface water, it dissipates at the surface of the lake (overflow or hypopycnal flow).

2. If the current is denser than the lake surface but less dense than the lake bottom water, it follows the thermocline (interflow or mesopycnal flow).

3. If the current is denser than the lake bottom water, it flows along the lake bottom (underflow or hyperpycnal flow).

One may add a fourth case:

4. If the lake is not stratified and the incoming current has the same density as the lake water, then the sediment dissipates within the entire water column, which then becomes homogeneously turbid (homopycnal flow).

In case 1 (overflow), no specific deposits are distinguishable within the sedimentary infill. These inputs become part of the so-called "continuous (or background) sedimentation", even if they are episodic and drastically vary through time. In contrast, cases 2 (interflows) and 3 (underflows) are the classic mechanisms behind the so-called "flood deposits" (often referred to as hyperpycnites for case 3), which are probably some of the most common event deposits encountered in lake sediments. The deposits formed under case 4 (homopycnal flow) may be indistinguishable when they represent the majority of clastic sedimentation, e.g., in the case of constantly turbid lakes, or they may be visible in the sediment core when they occur in generally clear-water lakes [35].

For all types of flood-triggered event deposits, the energy that transports the sediment within the lake basin is provided by the water injected by the tributary, driving the in-lake currents. In the case of interflow, this velocity is generally weak and rapidly decreases when the river enters the lake [34]. Such currents have a very low competence and transport only fine particles. However, the absence of friction allows these currents to propagate far within the lake [36]. In the case of large lakes, it has been shown that interflows are subject to the Coriolis force and are displaced toward the right in the Northern Hemisphere [36] and toward the left in the Southern Hemisphere. This effect is stronger at higher latitudes, where the Coriolis force is stronger. Sediment settling from interflows generally results in

the deposition of fuzzy silt layers across most of the lake [34]. In larger water bodies, these deposits may be concentrated on one side of the lake, depending on the hemisphere [36–38].

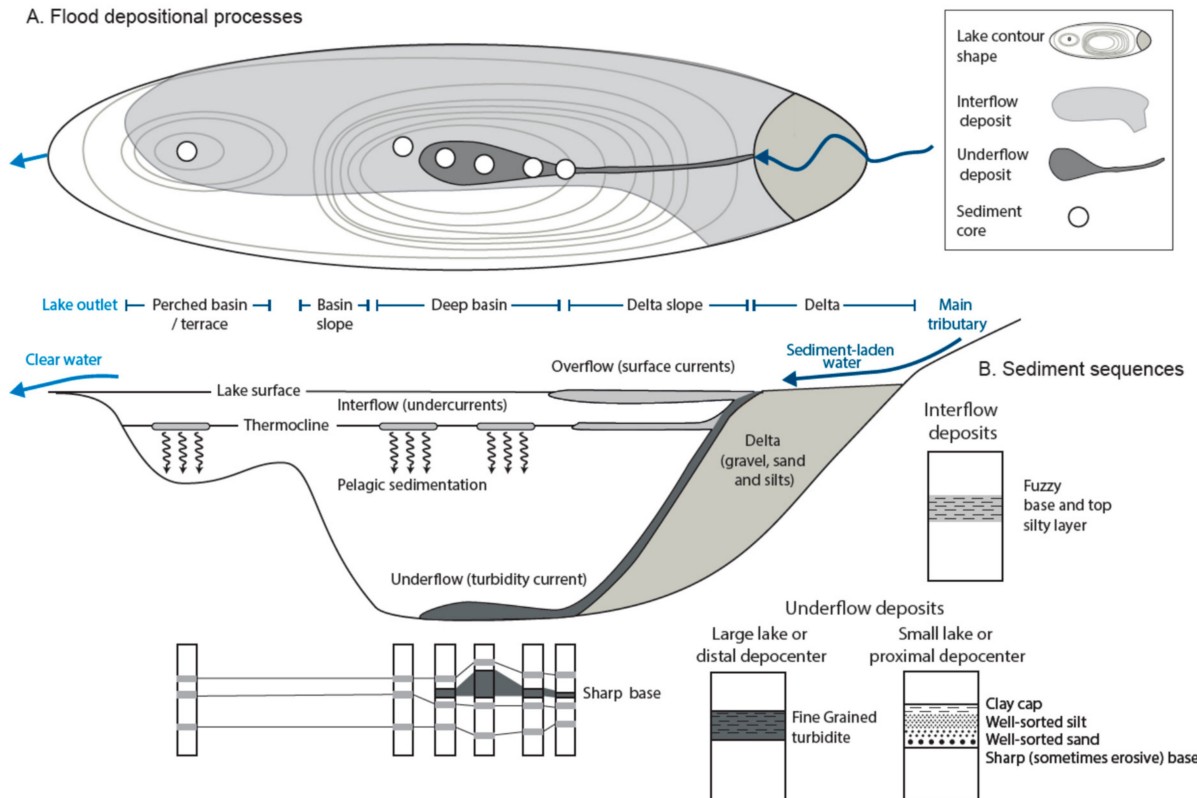

**Figure 2.** (**A**): Schematic representation of the processes occurring when a sediment-laden river enters a lake, with indication of the resulting sedimentary facies. (**Upper panel**): aerial view, (**middle panel**): transversal view (modified after [34,39]). (**B**): Typical sedimentary facies and locations of correlated interflow and underflow deposits, e.g., [36,40]. White dots represent cores; dark and light grey areas indicate underflow and interflow deposits, respectively.

Underflows occur during major, and thus highly erosive, flood events. In such a case, the stream or river enters the lake at a very high velocity, and its competence is maximal. Such a water mass is highly loaded with a wide range of particle sizes, making it denser than the lake bottom water, resulting in a current that flows along the lake bottom. It has been shown that the velocity of underflows correlates with the velocity measured in the stream itself [41–43]. The current velocity is hence sustained by water discharge and stops only when the discharge drops below a certain threshold. Such currents are highly turbulent, especially at their interface with the lake bottom. However, this turbulence tends to dissipation of the underflow energy, and the current velocity decreases along its journey within the lake basin [42]. These characteristics influence the shape and distribution of the associated deposits. Underflows follow the line of the maximum slope. As a consequence, underflow deposits are in general narrowly localized and present lateral facies variations from the most proximal (close to delta) to the most distal part [39,44]. In proximal lake environments, the high velocity makes the current erosive. Here, only the coarsest material settles and forms a layer with a sharp and erosive base [34,41]. The thickest part of the deposit occurs further along the underflow path, where it is no longer erosive [39]. Finally, as the flow becomes less turbulent, the friction at the lake bottom interface decreases, and the distal part of an underflow deposit may be identified far from the delta as a relatively fine-grained turbidite [36] but mainly in the deepest basin of the lake (Figure 2).

The two main types of flood deposits in lake sediments are, therefore, (1) fuzzy silt layers at locations influenced by interflows and (2) graded turbidites, with an average

grain size that decreases with an increasing distance from the river mouth until the deepest lake basin. Flood-triggered deposits can be characterized by a coarsening-upwards basal sequence formed during the rising part of the flood (waxing flow) and a fining-upwards upper sequence formed during the falling part of the flood (waning flow), named hyper-pycnite. However, the lowermost layers are often not preserved due to erosion during peak flow conditions [45]. Apart from a waxing flow, floods also typically last for hours to days and, therefore, often have a maintained peak flow, resulting in a very gradual fining trend in the deposit [18]. Such event deposits are best identified by grain-size analysis and multicoring since their spatial distribution may allow differentiation from other types of event deposits [40,46]. For more detail on flood-related event deposits, we refer to another paper in this special issue [9].

In contrast to flood-related deposits in lakes, only a few studies have focused on the lacustrine imprint of debris flows [47–49]. Recent work focuses on process-oriented investigations that involve both terrestrial and lacustrine geomorphological mapping and lacustrine sediment core analysis [49]. It appears that large parts of debris flows that enter the lake are deposited as lobes on steep alluvial fan slopes, whereas other parts evolve into turbidity currents. The associated turbidites in distal basin depocenters bear many similarities to flood deposits.

### 2.2. Glacial Lake Outburst Floods

Glacial lake outburst floods (GLOFs), also known as jökulhlaups, are catastrophic flooding events that occur when a lake dammed by a glacier or a moraine suddenly empties. They are relatively short-lived (<24 h) but can be very destructive, especially near the source lake. Compared to meteorological floods, their hydrographs start more abruptly and typically attenuate downstream. GLOFs originating from ice-dammed lakes are generally characterized by lower peak discharges than those from moraine-dammed lakes, as the enlargement of ice tunnels is a slower process than the overtopping and incision of moraines. Due to their abrupt nature, GLOFs generally rework large amounts of glacial material from the source lake and near the river headwaters.

Very few studies have investigated the sedimentary signature of GLOFs in lake sediments. In lakes located immediately downstream of the source lake, GLOFs are recorded as sand layers embedded in a finer-grained background matrix, most likely deposited under the influence of turbidity currents [10]. In lakes located more than 5 km away from the source lake, or in the presence of intermediate lakes, GLOF deposits typically correspond to thin and very fine-grained deposits (Figure 3), which are generally finer than background sediments [50]. Similar fine-grained characteristics have been described in fjord sediments, where GLOF deposits have a typical low total organic carbon (TOC) content [21]. Although these characteristics may be counterintuitive given the high hydrodynamic energy of GLOFs, they accurately reflect the glacial nature of the sediment transported by outburst floods, with typical mean grain-size values of ~5 μm and TOC values of ~0.2% (Figure 3). GLOF evidence has also been found in lakes sheltered from the proglacial river, where the most voluminous events caused the river to overspill, resulting in the deposition of fining-upwards and organic-poor slackwater deposits [51]. In proglacial Skilak Lake, Alaska, thicker varves were tentatively attributed to GLOFs [52]. The sedimentary signature of GLOFs in lakes with calving glaciers and, therefore, subglacial GLOFs is currently unknown, but fjord sediments indicate that such GLOF deposits may be very difficult to distinguish from background sediments, given their similar composition (e.g., [53]).

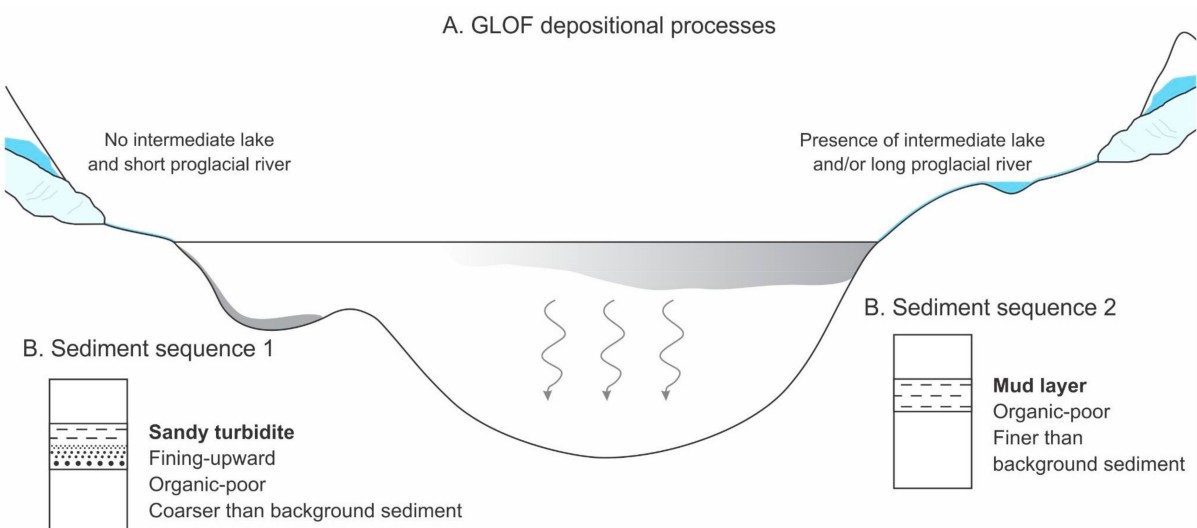

**Figure 3.** (**A**): GLOF depositional processes in lakes with (right) and without (left) intermediate lakes. (**B**): Sediment sequences associated with GLOF deposits.

The two main parameters used to identify GLOF deposits in lake sediments are, therefore, grain size and TOC. Although the grain size depends on the exact setting of the lake and distance from the source lake, TOC is consistently low due to the glacial nature of the sediment. The identification of GLOF deposits from lake sediment requires sufficient contrast between GLOF deposits and background sediments, which may be challenging in purely glacial lakes. This contrast is enhanced in lakes located sufficiently downstream, in which background sedimentation is generally more organic than the GLOF deposits.

### 2.3. Avalanches

Similar to the GLOFs, very few studies exist for deposits of snow avalanches. Pioneering work on this process has been carried out in Norway in postglacial colluvium [54]; however, if the avalanche path reaches a lake, snow avalanches can be recorded in lake sediments and by means of the debris they carry [55–57]. These studies indicate that avalanche deposits mainly correspond to coarse to very coarse minerogenic particles and macroscopic plant remains.

Most of the existing papers focus on wet avalanches, which are typically observed in steep alpine valleys predominantly during spring and are considered dense flows characterized by wet snow with a relatively high water content [58]. They are able to transport material ranging from fine soil particles up to gravel or boulders with terrestrial organic matter [56,59]. These wet avalanches are more effective erosional agents than winter dry snow avalanches and can entrain basal material. They thus act as important agents of sediment transport in steep alpine catchments [60]. Meteorological conditions from days to several weeks resulting in (1) the loss of strength by water infiltration, (2) overload of the wet snowpack by precipitation, and (3) gradual warming of the snowpack to 0 °C control the snowpack destabilization and wet avalanche triggering [61]. Even if the occurrence of wet avalanches is associated with positive snowpack anomalies [62], environmental parameters such as vegetation cover influence avalanche triggers. For instance, Fouinat et al. [12] highlight both the role of climate parameters (temperature and winter precipitation) and deforestation in past wet avalanche hazards.

Once an avalanche occurs, sediment is carried downslope by rapidly flowing water-saturated snow and can be integrated into lacustrine sediments in two ways depending on whether the lake surface is frozen or not [63]. In the case of a frozen lake, wet avalanche deposits are restricted to the lake surface close to the avalanche corridor, after which they spread across a large part of the lake by drifting ice during melting and drop to the lake floor (Figure 4). If an avalanche occurs while the lake is ice-free or possibly breaks the ice

when it hits the lake, the avalanches directly enter into the water, and then particles are concentrated in a more restricted area closer to the avalanche corridor (Figure 4). Given that wet avalanche deposits can be a local phenomenon, the coring location must be close to the avalanche corridor to capture both dropstones and direct avalanches; however, coring tends to be complicated by the presence of gravel at these proximal locations.

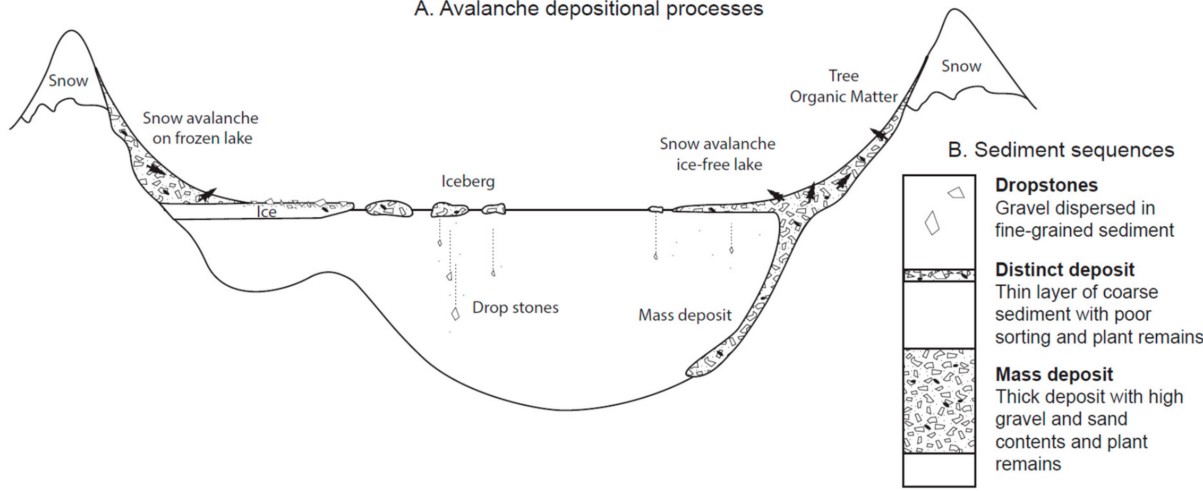

**Figure 4.** (**A**): Avalanche depositional processes on frozen (left) and ice-free (right) lakes. (**B**): Sediment sequences associated with avalanche deposits.

Direct wet avalanche deposits are recorded in lake sediment as multiple gravel elements at the same sediment depth in a fine, disturbed matrix, with very poor sorting. Dropstones induced by wet avalanches on a frozen lake surface can be recognized by a low number of gravel clasts in the fine-grained (laminated) background sediments [55,57,59,64]. Spatial coring surveys in the deeper lake basin allow us to better understand these depositional processes [64]. CT-scan methodology that allows the precise identification of coarse material embedded within the finer continuous sedimentation in a lake has been shown to be very useful for identifying past wet avalanche events [12,64].

### 2.4. Volcanic Eruptions

Volcanic eruptions produce large amounts of material that is eventually deposited on land and in aquatic environments. Although each eruption is unique in terms of the size, style, and composition of erupted material, the most explosive eruptions, i.e., those producing tephra, are those from felsic magma. Mafic magma, on the other hand, mostly produces effusive eruptions, i.e., lava flows, which are more local and have little influence on lake systems. Volcanic products emitted during explosive eruptions can reach several kilometers in the atmosphere and be transported over thousands of kilometers downwind. As such, tephra layers preserved in lake sediments are frequently used to reconstruct past volcanic activity, including the eruption frequency and magnitude, volume of erupted pyroclastic material, and geochemical evolution of volcanoes. They are more attractive than those recorded in subaerial terrestrial environments, in which erosion and chemical alteration often limit reconstructions to very large eruptions. Tephra deposits in lake sediments serve in tephrostratigraphy as isochronous features and in tephrochronology as very precise time markers (e.g., [65–67]). In addition, volcanic deposits can have pervasive ecological effects in lakes, such as changes in the ecology of aquatic communities [68–71].

Volcanic deposits in lake sediments can be grouped into two categories: (a) direct ash fall and (b) reworked deposits, which include lahars (Figure 5). Direct ash falls are encountered in lakes located immediately downwind of the eruptive center. These deposits can be recognized by their coarse and multilayered nature, representing the different phases of the eruption. They are generally coarser than background sediments and frequently

show one or several fining-upwards sequences, with the exception of pumice, which is the last type of particle to settle due to its low density. Pumice only sinks after water saturation, which typically occurs after a few weeks, depending on the grain size and vesicularity [72]. Therefore, direct ash fall deposits generally have a constant thickness across a lake, but pumice tends to be concentrated in the downwind direction and/or towards the lake outflow. Direct ash fall deposits, especially in small closed-basin lakes, are particularly suited to reconstruct erupted volumes since their thickness is representative of the magnitude of the eruption, e.g., [73].

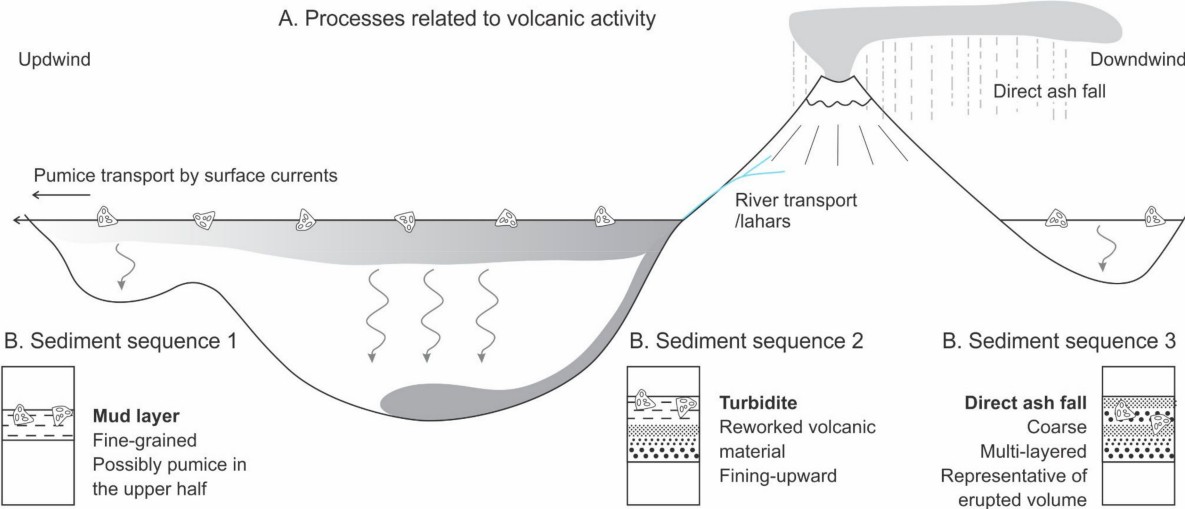

**Figure 5.** (**A**): Depositional processes related to volcanic activity: transport by surface currents (left), turbidites (middle), and direct ash fall (right). (**B**): Sediment sequences associated with deposits of different types of volcanic inputs.

Reworked volcaniclastic deposits are often found in large lakes with well-developed drainage systems (Figure 5). They also occur in lakes that were not affected by direct ash fall [74]. In that case, they provide evidence of river transport of direct ash fall deposits occurring in a specific part of the watershed. The volcanic material can be reworked immediately after the eruption or several years later by particularly erosive events, making these deposits less useful in volcanology than those from direct ash fall. The term "lahar" generally refers to a similar reworking of volcanic deposits by water but in which particles are particularly concentrated (debris flow). Upon entering a lake, volcanic material reworked by rivers generally forms turbidity currents, resulting in the deposition of typical turbidites in proximal basins (Figure 5). Their thickness varies according to the lake bathymetry, catchment size, and catchment area affected by the ash fall and is, therefore, not representative of the erupted volume [15,74]. In distal basins, only the fine material transported by interflows is deposited through settling, resulting in the deposition of thin mud layers. Pumice can occur in the upper part of turbidites and mud layers since they generally settle together with the finest volcanic particles (Figure 5).

Tephras are generally visible by the naked eye due to changes in color, and they often display sharp peaks in magnetic susceptibility. Tephra color can be highly variable depending on the nature of the erupted magma (lighter for felsic and darker for mafic). The most distinctive criterion of volcanic deposits is the presence of glass shards. As such, their petrology should be analyzed before attributing a volcanic origin to any distinct layers. Inorganic geochemistry is also relevant for tephra screening in lake sediments [75]. Although cryptotephras do occur in lake sediments, their particles are very fine-grained and mixed with background sediments, rendering their identification relatively challenging (e.g., [76,77]).

### 2.5. Earthquakes

Large earthquakes with either an on- or offshore epicenter generate strong seismic ground motions that can leave direct and indirect sedimentary imprints in lake sedimentary sequences. After a few pioneer studies in the 1970–1990s [78–81], a steeply rising number of lake records (currently ~120; [13]) have been studied worldwide for use as "natural seismographs" [82] that allow reconstruction of earthquake recurrence patterns [13,83]. To generate a sedimentary imprint, the strength of ground shaking (e.g., seismic intensity) must exceed a site-specific threshold, which depends on the type of imprint and the site characteristics. In most studies, this threshold is constrained by tracking the sedimentary imprint of the most significant recent and historical earthquakes in the region and by evaluating which earthquakes were recorded in the sediments and which ones were not [84–87]. Moreover, the lacustrine imprint mapping of recent and historical earthquakes revealed a wide range of sedimentary structures and deposits that can be induced by earthquake shaking [88,89].

These different earthquake-induced imprints are represented in Figure 6 and can be classified as follows:

(1) In situ soft sediment deformation structures (SSDSs): these structures are a direct consequence of seismic ground motion and mostly develop directly at the sediment–water interface. These consist of liquefaction features and brittle or ductile deformation structures [90,91] and are often called "seismites". As these develop in situ, they obliterate part of the slowly accumulated background sequence and should, therefore, not be excised from age–depth models, in contrast to all other event deposits in this manuscript (see Section 4).

(2) Deposits related to subaqueous mass wasting: earthquake shaking can trigger underwater mass movements due to the detrimental effects of transient shear stress and the resulting increase in the pore water pressure on the stability of lacustrine sedimentary slope sequences. Subaqueous mass movements, such as slides, slumps, and debris flows, lead to mass-transport deposits (MTDs) at the foot of basin slopes [92]. Moreover, turbidity currents are produced by dilution of the moving mass, leading to turbidite deposits in the more distal parts of the basins. Large turbidites with a thick homogenous middle unit are often visualized on seismic profiles as ponding acoustically transparent bodies and are often called "megaturbidites" [93] or "homogenites" [94]. Recent work inferred that thin seismo-turbidites can also be induced by remobilizing only a thin veneer (few centimeters) of surficial slope sediments and thus are not necessarily associated with subaqueous landslides [20].

(3) Deposits related to earthquake-triggered seiche action: see Section 2.7.

(4) Deposits related to terrestrial processes: There is a multitude of earthquake-induced terrestrial processes that may leave an imprint in lacustrine sequences. This includes (i) the direct propagation of triggered landslides into the lake, typically producing devastating impulse waves (see Section 2.7); (ii) triggered landslides in the catchment may lead to a prolonged period of clastic sediment supply ("catchment response") to the lake due to fluvial or eolian erosion and transport of loose landslide debris [95–97]. Such a catchment effect can be identified as closely spaced flood deposits or a generally more clastic background sedimentation; and (iii) changes in the hydrology of the catchment by altering the spring activity and composition, affecting the nature of background sedimentation [98].

Because the most commonly observed earthquake imprints relate to subaqueous mass-wasting processes, one of the most diagnostic criteria to identify earthquake-related deposits is formed by their spatial imprint. Geophysical mapping of MTDs on high-resolution multibeam bathymetric data [99] and reflection seismic profiles [100] is crucial for tracking the full basin-wide imprint of mass-wasting processes (see Section 3.1). Moreover, multicore analysis of seismo-turbidites can allow tracing of their sedimentary sources, i.e., submerged sedimentary slopes. Other highly important parameters for attributing a seismic origin to turbidites concern their compositional nature, such as geochemistry and their grain-size evolution from base to top. Turbidites that show an in-lake composition originate

from (deep-water) hemipelagic slopes, which are typically stable under static conditions and require seismic shaking to become unstable. Turbidites with a clastic allochthonous composition may be related to seismic or aseismic delta collapses (Section 2.6) or river floods and debris flows (Section 2.1). With respect to grain size, it is observed that seismo-turbidites show a poor sorting and sharp fining upwards trend in their basal part, reflecting a short transport path and rapidly decreasing flow velocity typical for subaqueous mass movements and associated turbidity currents [18,88].

By tracking the sedimentary imprint of well-documented historical earthquakes throughout multiple basins, it is inferred that the type and size of the sedimentary imprint is largely controlled by the local seismic intensity, i.e., the strength of seismic ground motion at the lake [86,89,101]. This implies that an earthquake source below the lake may produce similar imprints to onshore earthquake sources [102]. Moreover, the type and the size of sedimentary imprint can also be influenced by the duration and frequency content of seismic ground motion [97,103]. In some cases, megathrust earthquakes (long duration and low frequency) facilitate the triggering of multiple, voluminous landslides and the generation of megaturbidites while intraplate earthquakes (short duration and high frequency) may mostly induce onshore landslides, surficial slope remobilization, and the generation of thinner turbidites.

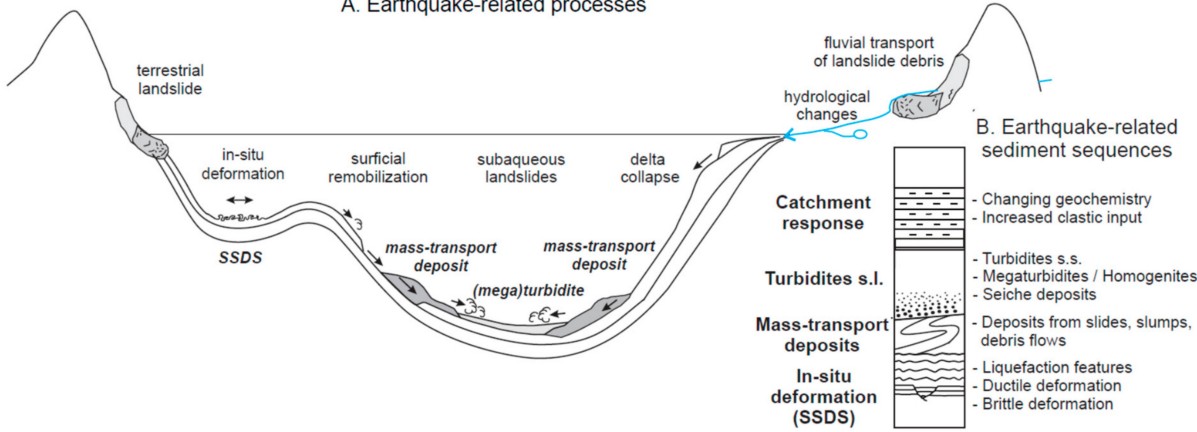

**Figure 6.** (**A**): Earthquake-related depositional processes through in situ deformation, surficial remobilization, subaqueous landslides, and delta collapse. (**B**): Sediment sequences associated with different types of earthquake-related deposits (modified after [89,104]). s.l.: sensu lato and s.s.: sensu stricto.

### 2.6. Delta Collapses

Delta collapses or "delta failures" refer to slope failures in (subaquatic) deltas. Delta collapses represent not only a major sediment transport mechanism from deltas towards deep basins but can also represent a major natural hazard, as they produce large mass movements and can be tsunamigenic, e.g., [105–107].

Historical delta collapses have been reported in Lake Brienz (Switzerland; [108]), Lake Geneva (Switzerland; [22,106]), Lake Lucerne (Switzerland; [105]), and Lake Kenai and Eklutna (Alaska, USA; [18,100]). These were recognized based on the thick sediment layers in the distal depositional areas of deltas (Figure 7). These distal deposits of delta collapses can be imaged on seismic reflection profiles as an onlapping seismic unit with low-amplitude to transparent seismic reflections, an erosive base horizon, and sometimes including blocks of high-amplitude reflections [105,106,108]. In some cases, large MTDs and megaturbidites were generated by delta collapses [100]. Due to high sedimentation rates in the deltaic areas, the source traces (e.g., scarps) of delta collapses are covered and obliterated very fast; thus, in historical case studies in Switzerland, the source of these delta collapses is not imaged on high-resolution bathymetric maps, e.g., [22,109].

**Figure 7.** (**A**): Depositional process of a delta collapse. The arrow in the deep basin shows the approximate location where the sedimentary sequence is expected. (**B**): The sedimentary sequence associated with a delta collapse (turbidite) compared to background sediments.

In sediment cores, the corresponding deposits are often described as (mega-)turbidites with a coarse sandy base and erosional features, such as incorporated mudclasts originating from the background sediments of the lake floor. The coarse-grained and erosive base is followed by a thick nearly homogeneous middle part with often a light-colored clay layer at the top, e.g., [105,108] (Figure 7). From these depositional patterns, it is assumed that the delta collapses generate a mass flow that develops into a turbidite. Whether the more homogeneous upper part of the deposit is the most terminal part of a turbidite or the consequence of a seiche needs to be further investigated.

The causes of these delta collapses are diverse and range from terrestrial mass movements impacting the delta plain [106] to floods [18], earthquakes [100,110], and sediment overload ("spontaneous") [105,108].

### 2.7. Tsunamis

Lakes can record both lake-internal and marine tsunamis. When tsunami waves reach the shoreline, the coastal zone is temporarily inundated during one or more uprush events (resulting from incoming waves). This inundation phase is followed by the backwash phase, during which water flows back to its source basin. Coastal lakes that are located within the inundation limit may thus record the uprush events of a marine tsunami (Figure 8). Similarly, lake tsunamis may be recorded in separate lakes or lagoons along its coast. However, lake tsunamis can also be recorded within the lake itself by backwash deposits or sedimentary signatures of currents related to the tsunami wave (Figure 8). Here, we first review lake tsunamis, followed by a review of how coastal lakes record marine tsunamis.

### 2.7.1. Lake Tsunamis

A variety of processes can trigger lake tsunamis, such as fault displacements, underwater volcanic eruptions, rock falls (causing impact waves), and mass movements, with the latter being the most common [14]. Studies of deposits resulting from lake tsunamis are much rarer than those of marine tsunamis and mainly exist for lakes in populated areas where the hazard of such lake tsunamis poses the highest risk, e.g., [106]. The deposits described in these studies are very diverse. In Lake Tahoe, a channel system with sediment waves and boulder ridges has been related to the strong backwash currents due to a lake tsunami [111,112]. In the nearshore (shallow) subbasins—that are distal from the tsunami source—of Swiss and Norwegian lakes, researchers identified graded deposits with a sandy (but up to pebble size) and sometimes erosional base, often including an organic-rich bed with plant debris and a fine-grained cap. These deposits were attributed to backwash [107,113] and/or erosion of nearshore sediments [113,114]. In all these studies, numerical tsunami modeling significantly aided the interpretation of the deposits.

A specific type of lake tsunami is a seiche (Figure 8), which—sensu stricto—is a standing wave that can evolve from a tsunami or be induced by tilting of the lake basin,

seismic waves, or wind [105,115–117]. When a seiche is produced by either of the latter two processes, the strong currents, especially in the shallow, nearshore areas, can be erosional. Currents can then transport this littoral material to the lake's depocenter, resulting in organic-rich intercalations [104] or "homogenites" in the base of which variations in grain-size sorting or anisotropy of magnetic susceptibility (AMS) provide evidence for oscillating currents during the seiche [30,116]. The seiche-induced currents are further thought to be responsible for the ponding geometry of these homogenites [88].

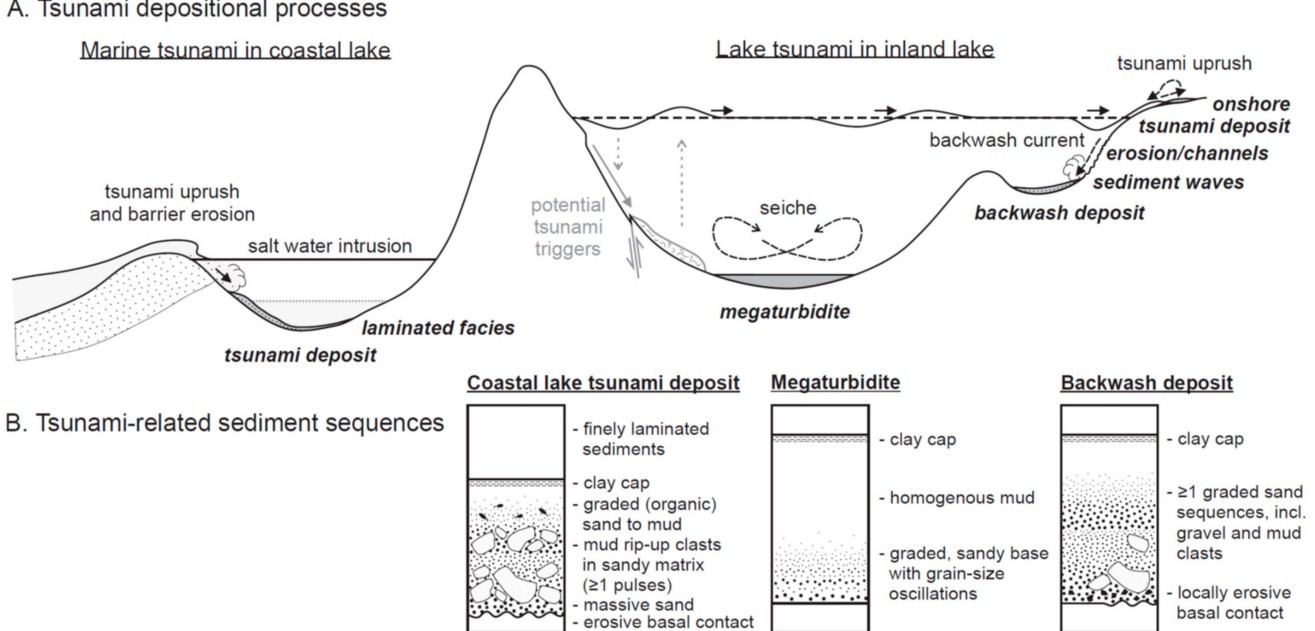

**Figure 8.** (**A**): Depositional processes of marine tsunamis in coastal lakes and lake tsunamis in inland lakes. (**B**): The different sedimentary sequences associated with tsunamis in lake systems.

Similar deposits can form when subaquatic or subaerial mass movements trigger the tsunami and a subsequent seiche [89,100,118,119]. These deposits usually have a graded, sandy base covered by a graded to homogenous mud that is described by some authors as a homogenite, and grain-size oscillations near the base of the muddy interval have been attributed to the seiche currents (Figure 8). The entire deposit (i.e., the sandy base and homogenous mud) is often referred to as a "megaturbidite", especially when it has a ponded geometry and can be distinguished on very-high-resolution reflection seismic data (Figure 8) [89,93,105]. However, megaturbidites can also be related to delta collapses (see Section 2.6) or be the final/distal deposit of mass movements, and the potential causal relationship with a lake seiche requires additional research. Whether the sediment showing evidence of the alternating currents in these megaturbidites is primarily derived from the initial mass movements or rather eroded by the seiche currents themselves [89,120] is not entirely understood and likely depends on the specific setting.

2.7.2. Marine Tsunamis

It is not surprising that the vast majority of coastal lake paleotsunami studies are along subduction zones, which are infamous for triggering the most destructive and widespread tsunamis during megathrust earthquakes. The first studies appeared in the nineties from Japan [121] and Vancouver Island [122], and more recent examples include coastal lakes along the Japan-Nankai [123], Japan-Honshu [116], Cascadia [124], Chile [125], Lesser Antilles [16,126] and eastern Mediterranean [127,128] subduction zones, and in Sri Lanka from Indian Ocean tsunamis [129]. Apart from such subduction zone tsunamis, the Storegga Slide tsunami has also been recorded in Norwegian coastal lakes [130,131].

The relatively limited number of coastal lake tsunami records is probably due to the specific conditions that need to be fulfilled in terms of the width, height, and geomorphology of the barrier separating the lake from the coastline, and the altitude of the lake. The lake should be sufficiently separated from the coastline such that only extreme wave events can reach the lake, but the altitude and distance should be sufficiently low for destructive tsunamis to inundate the coastal lake. Moreover, these conditions may change over time due to relative sea level variations [124,130] or dynamics in the barrier [132], resulting in a change in the sensitivity of a certain lake basin over time. In this respect, the outflowing river and related connection to the ocean have an important role, as they facilitate the uprush of the tsunami, thereby strongly influencing the effect of barrier dynamics over time.

The deposits of marine tsunamis in coastal lakes can be divided into two main categories (Figure 8): (1) deposits resulting from the high-energy uprush that erodes sediments in the nearshore area, barrier, and even in the lake itself; and (2) imprints resulting from salt water intrusion in a freshwater or brackish water system. Theoretically, another source of sediment could be the erosion of more inland areas and then deposition by backwash, as observed in marine bays [133].

(1) The high-energy deposits are typically fining-upwards deposits that can consist of, from bottom to top, an erosive and/or sharp basal contact, mud rip-up clasts in a sand/gravel matrix, and massive or graded beds ranging from sand (sometimes even gravel) to fine mud at the top, e.g., [124,125]. However, it also depends on the available material in the barrier. The spatial distribution of these deposits typically shows a strong gradient, with high-energy facies concentrated near the inflow of the tsunami and finer-grained basal facies away from the inflow [16,124,125,129]. The deposits regularly contain large organic remains; shell debris, e.g., [128,129,134]; and marine diatoms or foraminifera [16,121,124,134], the latter two of which are clear indicators of marine origin. In contrast, in a coastal lagoon, which is already brackish, the terrestrial (barrier) origin of the deposits may be the main indicator for a tsunami deposit [127]. The presence or absence of salt coating on sand grains, respectively indicating a barrier or lower-shoreface origin for sand grains, could also be used as a criterion to identify tsunami (reworking both barrier and shoreface sediments) from hurricane (mostly reworking barrier sediments) deposits [16]. Finally, different pulses of coarser-grained material within the general fining trend (e.g., layers with mud clasts or a sharp increase in grain size) have been attributed to separate uprush events and thus tsunami waves in several studies [125,130,131].

(2) The effect of salt water intrusion by tsunamis is often subtler but also of longer duration and is only present in freshwater lakes that do not have a tidal connection with the ocean. In such lakes, the dense salt water can form a lens of bottom water that does not mix with the fresh upper water layers, allowing an anoxic bottom layer to develop. Anoxia allows good preservation of fine laminations, which can last for several months to decades [121,124,135]. Kelsey et al. (2005) further demonstrated the presence of this salt water by the presence of brackish water diatoms in this interval.

*2.8. Cyclones, Hurricanes, Typhoons, and Storms*

Cyclones are large systems of winds that rotate around a center of low atmospheric pressure. Tropical cyclones, also named hurricanes or typhoons, are distinguished from extratropical storms on the basis of their driving parameters. Both are able to generate low pressures, very strong winds, large waves, storm surges, and abundant precipitation that can last a few hours to days and can lead to flooding of coastal areas. Storm surges can reach several meters and are responsible for major marine floods. In regions where the tidal range is large, marine floods occur when the storm surge is in phase with high tide and particularly during spring tide. Storm surges are produced by three physical processes: the inverse barometric effect, wind stress, and short waves (for more details, see the review of [11]). Three main flooding mechanisms can occur simultaneously during storm-induced flooding: overflowing or inundation, wave overtopping or overwashing,

and barrier breaching. Independently of the flooding mechanism, sediments from the barrier (potentially originating from the shoreface, the foreshore, and/or the aeolian dune) are transported in the backshore area. Three main types of sedimentary signatures of storm surges are found in backshore settings: washover fans, beach ridges, and discrete records deposited in coastal lagoons, coastal lakes, and sinkholes [11]. Here, we focus on sediment records in coastal lakes, enclosed lagoons, and sinkholes. Coastal lakes (or ponds) are water bodies enclosed in topographic depressions and separated from the sea by a barrier, most often sand-dominated [136,137]. Coastal lagoons can be enclosed or semi-enclosed with one or a few tidal inlet(s) [138,139]. Sinkholes and blueholes are particular cases of coastal lakes and originate from the collapse of the earth surface layer [140,141]. Karst formation and dissolution are dominant processes in sinkhole generation.

Pioneering works on storm sediment records in coastal lakes were conducted in Alabama, along the Gulf of Mexico, by Liu and Fearn (1993). These authors found multiple sand layers, 0.1 to 1 cm thick, interstratified in organic clay sediment. After using the stratigraphic signature of a recent and well-documented hurricane as a modern analog, these layers were interpreted as resulting from overwash of the beach and dunes (Figure 9).

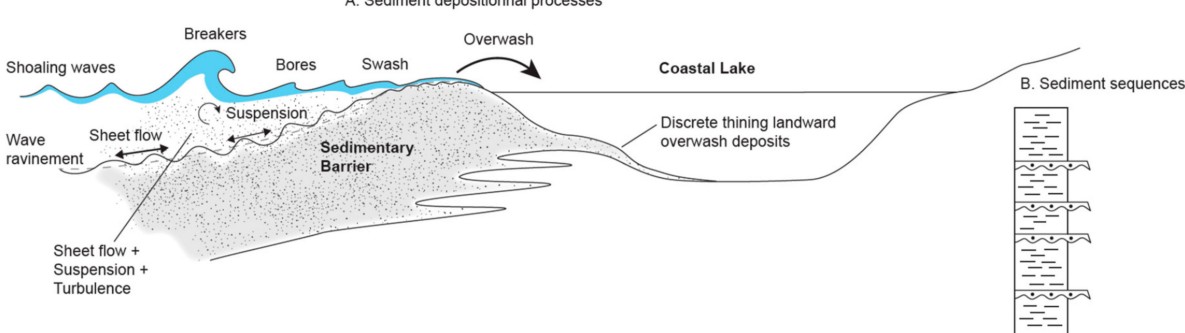

**Figure 9.** (**A**): Depositional process of cyclones and hurricanes. (**B**): The sedimentary sequence associated with cyclones and hurricanes compared to background sediments.

Sediment records of past hurricanes were evidenced in many coastal lakes and lagoons along the US coast in Alabama [142], Florida [136,143], and Massachusetts [144]. Similar records were found in coastal lakes and lagoons located in the Caribbean [16,138], in Japan [145], in China [146], and along the Mediterranean coastlines [139,147–151]. Famous examples of storm records were also found in coastal sinkholes in Florida [140,152] and in the Bahamas [153] and in blueholes in Belize [154] and in the great Bahama bank [141].

Hurricanes and storms recorded in coastal lakes, enclosed lagoons, and sinkholes are typically coarse-grained deposits, which contrasts with the normal low-energy fine organic sedimentation. They are frequently dominated by sand (Figure 9) but can also be identified by proxies of fine sediment such as geochemistry and clay mineralogy, e.g., [155]. This coarser composition is interpreted as recording an increase in energy due to intense wave activity and to an input of sand eroded from the barrier to the coastal lake. Modern analogs, consisting of historically well-known storm deposits, are frequently used for the interpretation of these coarse layers [16,139,140,144]. Most of the storm layers found in coastal lakes are typically a few millimeters to centimeters thick and thus can be defined as discrete deposits (Figure 9) in contrast with more massive storm deposits, such as washover fans and beach ridges [11]. Storm layers in coastal lakes are also frequently characterized by a lower content in organic matter with respect to normal sedimentation [156,157] and by inorganic geochemistry [145,155]. Storm layers frequently include marine shells, such as marine benthic foraminifera [140,158,159] or marine mollusks [139], which contrast with coastal lake or lagoonal species associated with normal sedimentation. For more details on storm proxies, see [160].

Hurricane sediment records were frequently used to reconstruct the recurrence interval of major storms and the storm history for the last millenaries. Such reconstruction usually shows alternations of periods with strong storm activity and more quiescent periods, which are related to climate changes [16,136,138,144–146,148,151].

### 3. How Can Event Deposits and Associated Processes Be Identified?

This section focuses on the spatial imprint, lithofacies, and biological and organic facies of event deposits. It discusses the main approaches and proxies (summarized in Table 1) that allow distinguishing the processes at the origin of event deposits in lakes. Basic criteria such as the color, magnetic susceptibility, and density of sediment cores may be useful to identify the presence of event deposits but are generally not sufficient to determine the type of event deposit and are thus not discussed in this review.

**Table 1.** Summary of the different types of approaches and proxies reported in three main groups (spatial imprint, lithofacies, biological and organic facies) allowing event deposit characterization. The relevance of different proxies for each event deposit is indicated as follows: **X** = main importance, x = secondary importance, - = minor importance.

| Event Deposit Type | Spatial Imprint | | | Lithofacies | | | | Biological and Organic Facies | |
|---|---|---|---|---|---|---|---|---|---|
| | Bathymetry | Seismic Reflection | Core Correlation | Grain Size | Sedimentary Structures Microfacies | Inorganic Geochemistry | Petrology | Organic Geochemistry | Biological Remains |
| Flood | x | x | **X** | **X** | x | **X** | x | x | - |
| GLOF | x | x | x | **X** | x | x | - | **X** | - |
| Avalanche | x | x | x | **X** | x | x | x | **X** | - |
| Cyclone/Hurricane | x | - | **X** | **X** | x | **X** | x | x | **X** |
| Volcanic eruption | x | x | x | x | x | x | x | **X** | - | - |
| Earthquake | **X** | **X** | **X** | **X** | x | x | - | x | - |
| Delta collapse | **X** | **X** | **X** | **X** | x | x | - | x | - |
| Marine tsunami | x | x | **X** | **X** | **X** | x | x | x | **X** |
| In-lake tsunami | **X** | **X** | **X** | **X** | **X** | - | - | x | - |

### 3.1. Spatial Imprint

#### 3.1.1. Bathymetry

A good bathymetric map is important not only for all limnogeological studies to select the appropriate coring sites for the study purposes but also to obtain the first ideas of the most recent sediment features, processes, and deposits. In many cases, the most complete sedimentary sequence can be found in the deepest part of the lake, which is best sheltered from erosion or nondeposition occurring in dynamic coastal environments or caused by lake level variations. Moreover, deeper parts often contain anoxic water bodies. The resulting lack of bioturbation facilitates the preservation of thin layers and annual laminations (varves), which represent a high-resolution dating tool (see also Section 4). Another important value of high-resolution bathymetric data acquired by multibeam bathymetric sonars is to identify and understand different types of event deposits (see Table 1). In particular, relatively recent MTDs and their corresponding subaqueous source areas (scarps) can be accurately mapped, and their potential triggers can be evaluated. For example, human-induced subaqueous MTDs may be caused by coastal infrastructure works (Figure 10A,B; [99,161]), whereas other MTDs could be traced back to delta collapses (Figure 10B; [92,162]) or earthquake-induced failure of hemipelagic slope sequences [161,163]. Moreover, high-resolution bathymetric data allow the identification of deposits related to terrestrial mass movements such as rockfalls, rockslides, and rock avalanches (Figure 10C [114,164–166]).

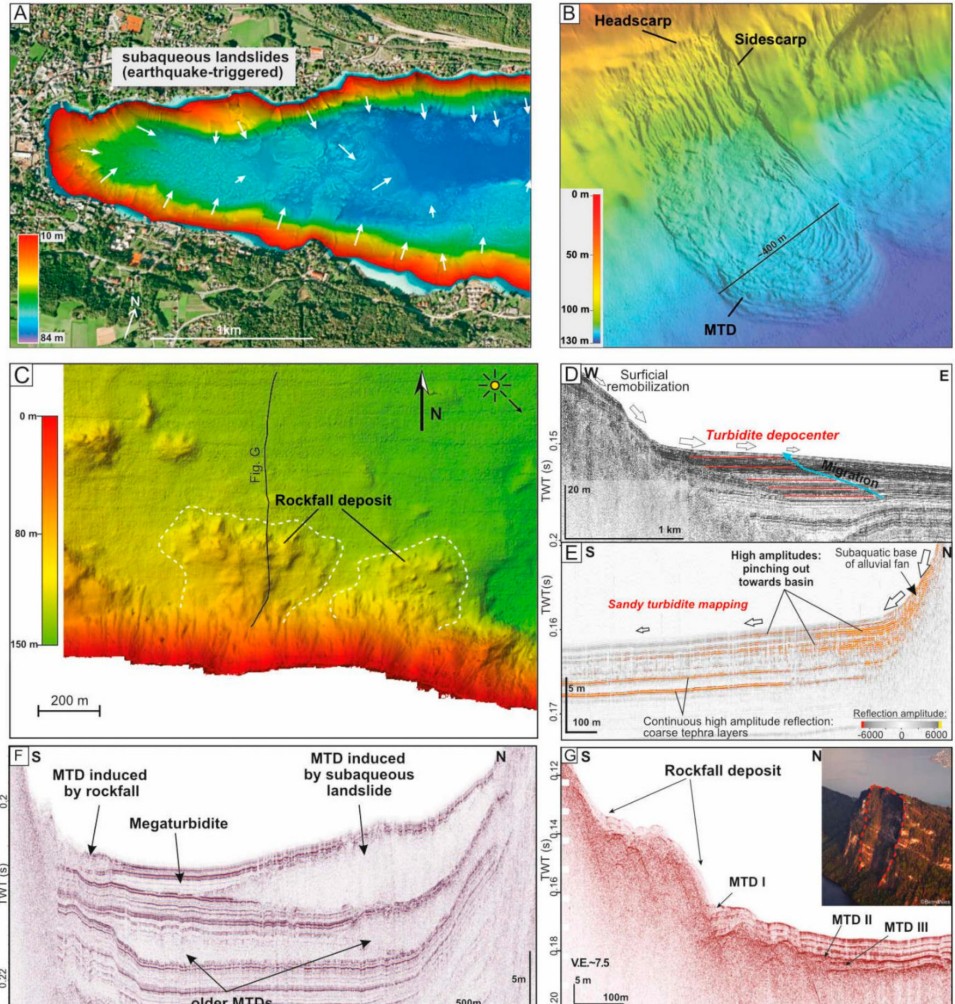

**Figure 10.** (**A**): Bathymetric map of Lake Woerthersee with indication of subaqueous mass movements (white arrows). Mass movements were identified by scarps in their source area and the positive morphology (blocks, ridges) of the corresponding MTDs (modified after [99]); (**B**): 3D perspective on the bathymetric expression of a subaqueous mass movement in Lake Zurich. Note the scarps in the evacuation area and the compressional ridges in the area (modified after [167]). (**C**): Bathymetric expression of two rockfall deposits in Lake Lucerne (modified after [164]) with the location of the seismic line in (**G**). (**D**): Seismic reflection profile in Lake Riñihue indicating the spatiotemporal migration of a turbidite depocenter. This depocenter is identified as a ponding geometry with onlapping reflection terminations (modified after [168]). (**E**): Subbottom profile in Lake Riñihue at the base of an alluvial fan, showing lateral changes in the reflection amplitude. These are related to the thinning and fining of sandy turbidites that originate on the alluvial fan (modified after [86]). (**F**): Subbottom profile in Lake Lucerne showing the seismic facies and geometry of a megaturbidite, an MTD induced by subaqueous landslides, and an MTD associated with the impact of a rockfall on the lake floor (modified after [119]). (**G**): Subbottom profile of a rockfall cone (presented in (**C**)) and associated MTDs in Lake Lucerne. Inset: source area of the rockfalls (modified after [119]).

Other types of geomorphological structures observable on high-resolution bathymetric data are active fault segments [102,169]. The typical resolution of state-of-the-art multibeam bathymetric data in moderate-depth (50–200 m) lakes is approximately 1–2 m horizontally and a few decimeters vertically.

### 3.1.2. Seismic Reflection

Event-related deposits and structures that are several 100–1000 years old are typically covered by a significant amount of sediment, and their traces may no longer be visible on lake bathymetric data. This may be extreme in high-sedimentation and dynamic delta areas, where a few years of sedimentation may suffice to obliterate evidence of significant gravitational mass movements or fault displacement. To obtain a temporal perspective on the occurrence of large event deposits, dense networks of very-high-resolution 2D reflection seismic profiles are acquired, and seismic-stratigraphic analysis is performed. The frequencies used in very-high-resolution seismic data typically range between 1 and 10 kHz [170]. Vertical resolution in the decimeter range is achieved and allows correlation of reflectors with small grain size changes and/or shell beds in homogenous sediments [171,172]. Seismic profiling allows the identification of (1) turbidite depocenters and the direction of clastic input, (2) individual megaturbidites, and (3) mass-transport deposits from subaqueous or terrestrial slope failures.

(1) Turbidite depocenters can be identified by a thickening of stratigraphic units, especially in the deeper parts of basins that are near the potential source area of turbidity currents such as river inflows or subaqueous sedimentary slopes. Seismic-stratigraphic mapping allows the identification of the possible migration of such depocenters over time (Figure 10D), which may be crucial for the interpretation of long sediment cores [22]. Moreover, sandy turbidites interbedded in muddy background sediments generate high-amplitude reflections, and thus, their extent can be traced from the base of the slope towards more distal areas (Figure 10E). A similar reasoning applies to the tracking of sandy marine tsunami deposits in coastal lakes from the point of tsunami inundation towards the more distal undisturbed lake parts (Section 2.7; [125,132]).

(2) The homogenous unit of megaturbidites can be identified as an intercalated transparent body that ponds in the deeper parts of a basin, partly leveling out previous morphological features. Given its acoustically transparent facies, it is often called homogenite in the literature. The coarse basal part of megaturbidite can be identified by high-amplitude reflections, and it thickens towards the source area of the reworked sediments [106] (Figure 10F).

(3) Mass-transport deposits form a wedge-shaped positive morphology with a chaotic-to-transparent acoustic facies and are directly located at the base of slopes (Figure 10 B). Muddy MTDs that originate from subaqueous failures exhibit more elongated shapes compared to rockfall-induced accumulations of debris (Figure 10E). Multiple MTDs on a single stratigraphic level point towards a synchronous failure of several slopes. Especially when this involves subaqueous slopes, this coeval MTD signature forms a strong argument to infer the occurrence of strong paleoearthquake shaking [173]. MTDs can be directly covered by a megaturbidite, which may be related to the dilution and flow transformation of the subaqueous landslide (Figure 10E) and—potentially—seiche effects (Section 2.7). As subaqueous landslides can significantly erode and deform basin-plain sediments [163], sequences in sediment cores may be overturned or duplicated [17].

### 3.1.3. Core Correlation

The spatial distribution of event deposits is one of the key characteristics that allows the determination of their source and trigger. To obtain the spatial distribution for thin deposits that cannot be traced by subbottom profiles, such as centimeter-scale turbidites and tephras, high-resolution core-to-core correlation is an important prerequisite. Several methods and strategies can be used to correlate cores within a lake basin. In the case of (finely) laminated sediments that exhibit marked color variations, visual core correlation can be achieved down to millimeter-scale accuracy (Figure 11A,B; [18,23,39,86]). Alternatively, (near-)continuous scanning or logging data, such as XRF, magnetic susceptibility, or X-ray computed tomography (CT), can be used to correlate gradual or abrupt variations through time of the background sediments (Figure 11C [19,102,174]). Additionally, independent (varve-counted) age models for each single core have allowed the correlation of cores

within a lake basin [175]. To support the correlation of background sediments, or when they are homogenous and do not allow high-resolution spatial correlation, marker beds can be used. Classical marker beds are air-fall tephra layers, which are spatially uniform event deposits within a lake basin and thus present an ideal time marker. If not macroscopically visible, tephra layers are often represented by a peak in magnetic susceptibility, and the pattern of these magnetic susceptibility peaks then allows for core correlation [67,176]. Geochemical fingerprinting of the glass shards can further confirm these correlations between cores within one lake [177] and can even permit core correlation between (nearby) lakes [89,178] or link the tephra layers with well-documented tephra deposits in outcrops or eruptions [67,179]. Finally, other event deposits could be directly correlated between cores based on their specific characteristics (Figure 11A: lahars; Figure 11: floods), although such correlations should be carried out with care and supported by independent methods, for example, absolute or relative dating [39]. The established core correlation can be used to determine relative differences in the sedimentation rates, which should be in line with the thickness variation of the corresponding sediment package as inferred from seismic stratigraphy (see Section 3.1.2 [176]), and thereby independently supports the core correlation. Near active faults that cross-cut the lake floor, differences in sedimentation rates from both sides of the fault segment have been used to estimate fault displacement [102].

An independent high-resolution core-to-core correlation is paramount for studying most event deposits, as it aids (1) the identification of the event deposits and (2) mapping of their spatial distribution.

(1) In rather homogenous sediments, core correlation facilitates the identification of event deposits, which often have a more heterogeneous spatial distribution compared to the background sediments. Abrupt differences in the thickness between correlated horizons or marker beds may thus indicate the presence of an event deposit ([86]; Figure 11A). Furthermore, core correlation is often the only method to identify erosion caused by the often more energetic hydrodynamic conditions during the event [18]; Figure 11B). Finally, in some cases, core correlation is paramount for distinguishing SSDSs from slumps, as the latter are intercalated, while SSDSs develop in in situ background sediments.

(2) The spatial distribution of event deposits often allows determination of their trigger because it is related to a combination of the sediment source location and the hydrodynamic processes. The usually decreasing energy with travel distance over the basin plain causes much sediment to be deposited near its source region (e.g., inflow, landslide), resulting in a heterogeneous distribution with thicker (and coarser-grained) deposits near the sediment source (Figure 11D). Known examples are tsunami deposits in coastal lakes, which are typically coarser and thicker near the dune barrier and/or river connection to the ocean [16,124,125]; backwash deposits of lake tsunamis are only present in the subbasin close to where the tsunami remobilized shore material [107]; lahar deposits are thickest near the inflows of specific lahar events [15]; gravel deposited by snow avalanches is focused near the avalanche pathways [59,64]; and turbidites related to spontaneous delta collapses are coarser and/or thicker near that delta [22]. The spatial distribution may aid in distinguishing flood- from earthquake-induced turbidites, a common challenge when interpreting lacustrine turbidites [88,176,180]. Floods can trigger underflows, causing localized deposits with lateral facies variations from the most proximal to the most distal part (Figure 11D; [39,44], see Section 2.1). The thickest part of the deposit occurs further along the underflow path, where it is no longer erosive [39], and the distal part of an underflow deposit may be identified far from the delta but only in the deepest basin of the lake [35,36]. Floods can also cause inter- and/or overflows (see Section 2.1 and Figure 2), which result in widespread deposits covering relatively shallow areas as long as these are located under the thermocline (in the case of interflows). Interflow-related turbidites in such shallow distal sites are thin and fine-grained (Figure 11E; [18,34]). In contrast, earthquake shaking can remobilize subaquatic slope sediments, and the associated turbidites thus show a more basin-focused distribution and are absent from shallow subbasins or platforms

(Figure 11E; [18,46]). Earthquake-induced turbidites also regularly show localized deposit accumulations near failed slopes [181].

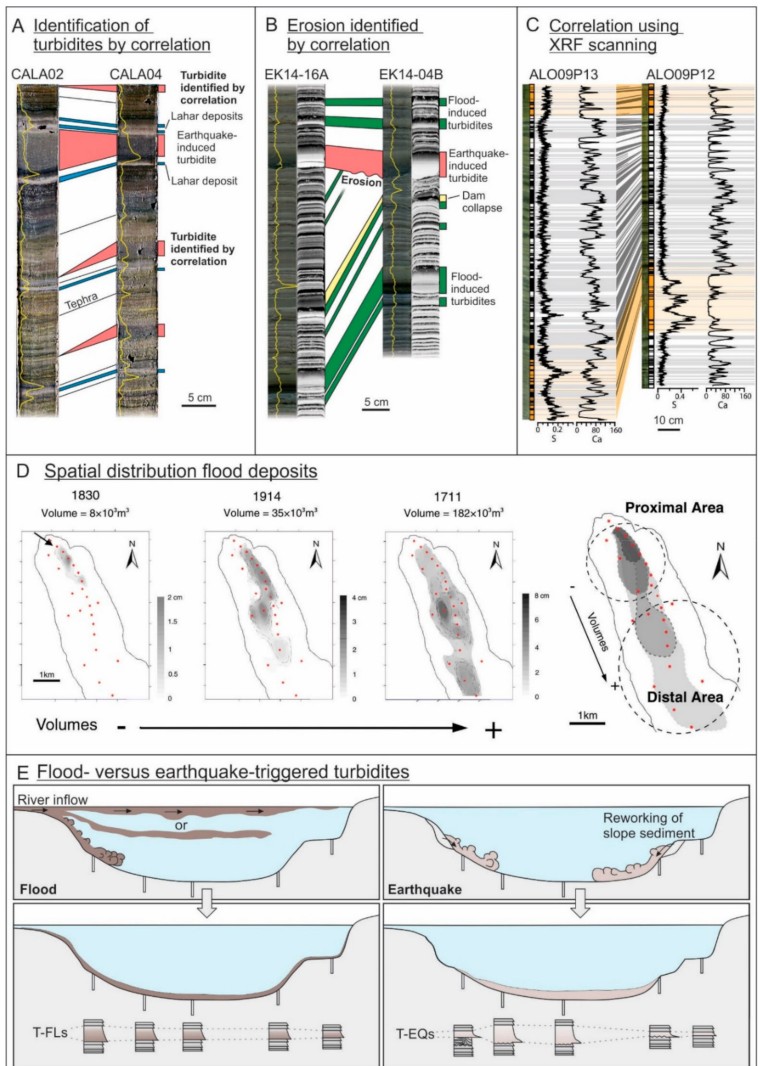

**Figure 11.** Examples of core-to-core correlations in lakes. (**A**): Core images with the magnetic susceptibility (yellow line) of sediment cores from Lake Calafquén (Chile; modified after [86]). Interpretation of some of the earthquake-induced turbidites (red) is aided by core correlation. More evenly distributed lahar deposits (blue) help correlation. (**B**): Core images (left) and X-ray CT images (right) of sediment cores in Eklutna Lake (Alaska; modified after [18]). The identification of erosion below the earthquake-induced turbidite (red) is much easier due to core correlation. Flood-induced turbidites (green) help correlation. (**C**): Core images (left), lithology (middle; white: mud; yellow: organic-rich mud; black: graded beds), and sulfur and calcium XRF counts (right) of sediment cores from Lake Allos (French Alps; modified after [182]). The high-resolution XRF data allow detailed correlation between cores. (**D**): Maps with distributions of flood deposits in Lake Le bourget (France; modified after [39]). Red dots indicate the core locations that have been used to produce the maps and calculate the volumes. Larger floods, which transport larger volumes of sediments, spread over wider, more distal areas of the lake. (**E**): Conceptual drawings of differences between processes and deposits related to floods and earthquakes in Eklutna Lake (Alaska; [18]). Core-to-core correlation shows a widespread distribution of flood deposits (T-Fls) due to interflows, albeit much thicker near the source river because of the influence of underflows. In contrast, earthquake-induced turbidites (T-EQs) have a much more basin-focused distribution because they result from sediment remobilization on subaquatic slopes.

The spatial extent and thickness of event deposits have additionally permitted researchers to estimate local intensities and ultimately the. magnitudes of prehistoric events. For example, using the cumulative turbidite thickness, Moernaut et al. (2014) were able to determine the macroseismic intensities of earthquakes, eventually leading to magnitude estimations of paleoearthquakes [183]. Similarly, the turbidite extent and thickness have been related to the intensity of lahars [15] and floods [17,35,39,184], resulting in quantitative event reconstructions.

Having multiple strategically located cores from a single event deposit thus significantly expands the amount of information that can be obtained from the deposits by facilitating the determination of not only the trigger mechanism but also sometimes the intensity of the event. Furthermore, in some cases, multiple cores are also required to obtain a complete event stratigraphy, as local erosion may hamper record continuity and many event deposits have a heterogeneous distribution in the lake basin.

### 3.2. Lithofacies

#### 3.2.1. Grain Size

Grain size is one of the most fundamental physical properties of sediments. It is widely used as an indicator of hydrodynamic conditions and is, therefore, one of the best proxies for reconstructing transport processes responsible for event deposits. It is most commonly measured using laser-diffraction particle size analysis, which is generally limited to a downcore resolution of ~5 mm due to the need for subsampling. A way to overcome this issue consists of predicting the sediment grain size using other high-resolution measurements, such as inorganic geochemistry [185], $\mu$CT scans [59,64,186], and hyperspectral imaging [187]. For example, millimeter-scale variations in grain size can be achieved using ratios of elements measured by XRF core scanning (Figure 12A; [19]) because sediment inorganic geochemistry reflects the mineralogical composition, which is grain-size dependent [188]. One other main limitation of laser-diffraction particle size analysis is that it is limited to particles finer than 1–2 mm. Samples containing coarser particles are generally combined with dry or wet sieving (e.g., [94]).

Grain-size data have traditionally been reported using the statistical parameters of distributions such as mode, mean, median (D50), size of the coarsest fractions (D90, D99), sorting (sigma), skewness (Sk), and kurtosis (K) [189]. A more complete way to represent grain-size distributions consists of contour plots or heatmaps, which represent the wholedown core distribution (Figure 12B). Finally, grain-size distributions can also be decomposed into mixtures of subpopulations. This is particularly suited to multimodal distributions, which suggest the co-occurrence of different sedimentary processes. These populations may be identified by a simple standard deviation method to detect the grain-size intervals with the highest variability [139] or extracted from parametric [190] or nonparametric end-member modeling with, for instance, the EMMAgeo [191] and AnalySize [192] packages or the QGrain software [193].

For the identification of event deposits, the most interesting approach is generally to study the evolution of grain-size parameters with depth. A typical example is grading in turbidites, which represent a transient flow. However, some event deposits do not show any grading and are characterized by homogeneous facies covered by a clay cap. Such deposits are often referred to as homogenites and can be related to a seiche effect (see Section 2.7; [116]) or can be an indication of en masse deposition, especially in the case of the additional presence of coarse grains in a muddy matrix [194]. In an influential article, Passega [195] proposed that the depositional characteristics of a process are reflected in the texture of the sediment and can be summarized by two parameters of the grain-size distribution: D50 and D90. Since then, the Passega biplot has been frequently used to analyze transport mechanisms. We also recommend the use of biplots with the thickness and D90 max (i.e., the highest D90) of each interbedded event deposit [17,182].

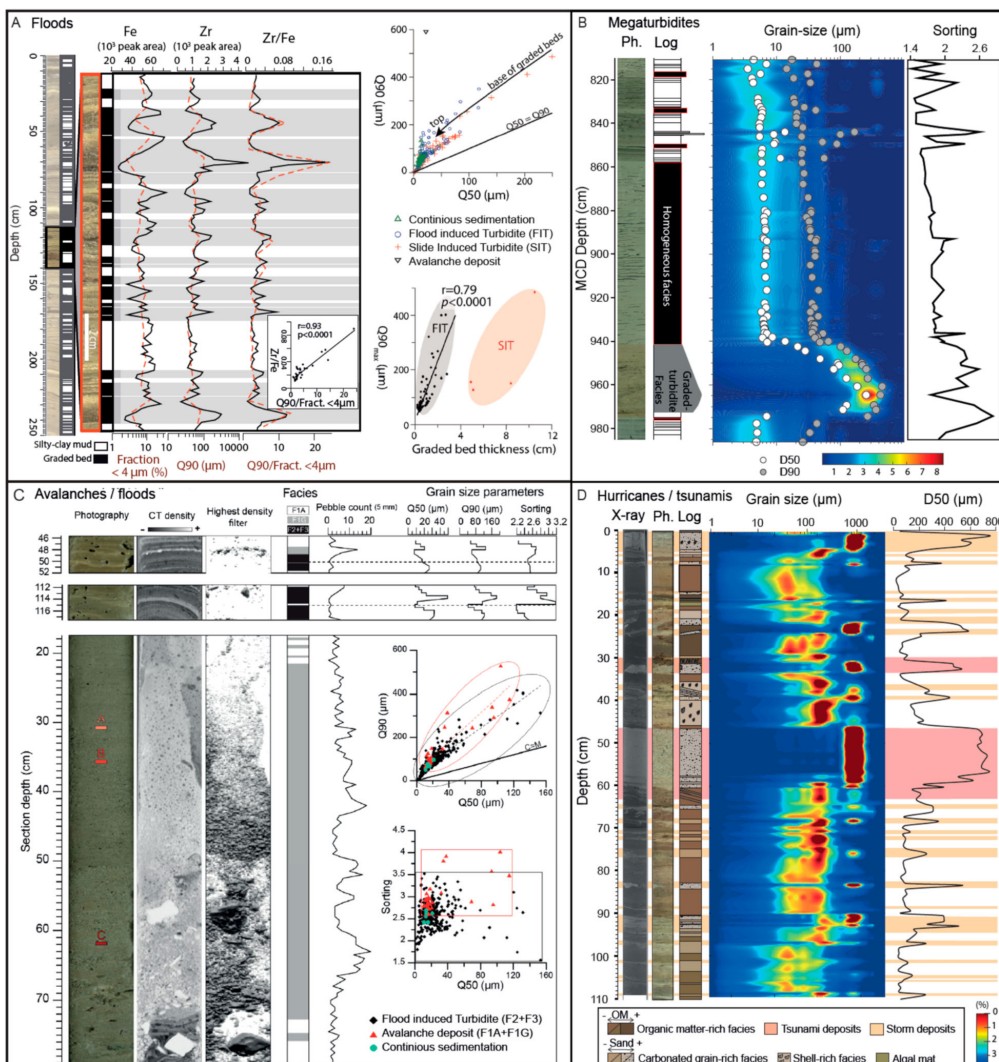

**Figure 12.** (**A**): Grain-size and geochemical data for avalanche- and flood- and slide-induced turbidites in Lake Blanc des Aiguilles Rouges [19]; (**B**): Grain-size data for megaturbidite with a graded layer and then a homogenite facies from Lake Aiguebelette (Banjan et al., submitted); (**C**): Grain-size and CT-scan data for avalanches and flood deposits in Lake Lauvitel [12], in this subfigure A,B,C are location of individual grain size data see [12] for more details; (**D**): Grain-size data for storm and tsunami deposits in a small coastal lagoon on Scrub Island [16].

As different types of event deposits can co-occur in the same sediment archive and as distinct events of one type can result in a similar general grain-size signature (e.g., normally graded turbidite), the grain-size parameters and thickness are a good way to distinguish them (Figure 12). For instance, flood turbidites typically present a fining-upwards trend but may be too thin to fully observe grain-size variations with depth. However, most flood turbidites present a good positive correlation between some grain-size parameters, such as Q90 or Q50, and the deposit thickness while for a given grain size, earthquake-induced deposits are thicker, and for a given thickness, avalanche deposits are coarser (Figure 12A; [17,19]). Globally, the D90 vs. D50 relations are different for various kinds of event deposits (Figure 12A,C). In the case of flood deposits, grain size can be used as a flood intensity proxy because it reflects the transport capacity and ultimately the precipitation intensity [35,196,197]. In some lake systems, if a relation between Q90max and thickness is well defined, this last parameter could be used as an intensity proxy [17,19]; for more detail, see Wilhelm et al. in this issue [9]. In lake systems, the theoretical basal reverse grading in flood deposits (hyperpycnite) due to initial waxing of the underflow [198] is rarely

preserved due to the increasing flow velocity, which erodes or prevents the deposition of fine-grained basal deposits. However, the gradual waning of the underflow following the waxing leg is often preserved and represented by a very gradual grading of the turbidites, as opposed to steep grading in turbidites related to slope failures (such as earthquake-induced turbidites) [18].

Grain size can also be useful to attribute a seismic origin to turbidites, megaturbidites, and homogenites [82,87–89]. As discussed above, seismically induced turbidites may have different grain-size/thickness parameters than flood-induced turbidites (Figure 12A). In megaturbidites (Section 2.7), the bed-load sediment generally produces a (thin) coarse base, whereas the suspended sediment is homogenized by the oscillation of the lake waters by the seiche effect, resulting in very stable grain-size parameters (Figure 12B; [17,30,116]). The grain size is also an ideal proxy to define avalanche deposits in lake systems, as these layers are mainly characterized by a heterogeneous distribution with coarse to very coarse mineralogical particles and very poor sorting (Figure 12C;). In such cases, CT scans are a very useful tool to detect, measure, and count single coarse grains [12,64]. The grain size is also widely used to identify and characterize sandy deposits related to storms and hurricanes [16,138,139,141,144] or tsunami deposits [16,125,199] in muddy organic-rich lagoons or lake sediments (Figure 12D). However, distinguishing storms from tsunami deposits remains challenging in coastal systems and seems to be site specific [200]. If a sedimentary sequence presents both types of deposits and tsunamis may be regarded as a more energetic process, the grain size is not necessarily relevant for both processes because it primarily depends on the same available material from the sandy barrier (Figure 12D). However, grain orientation, petrology, and sedimentary features such as rip-up clasts could help [16].

### 3.2.2. Sedimentary Structures and Microfacies

Sedimentary structures are common in event deposits due to the often energetic conditions that occur during deposition. Whereas sedimentary structures can be observed on split core surfaces, the orientation of the core surface relative to the orientation (strike, dip, vergence, etc.) of the structure strongly determines its expression. Hence, CT scanning is becoming a popular tool to image lake cores, as it allows 3D visualization of sedimentary structures. Medical CT scanners commonly reach a resolution of a few hundred micrometers, which is sufficient to identify sedimentary structures as long as there is a contrast in radiodensity (depending on the density and atomic number). Lab-based μCT scanners allow higher resolutions down to a few dozen micrometers or even lower, although such higher resolution comes with the drawback of a smaller sample size (~1000–2000 times the resolution, i.e., 1–2 cm for a 10-μm resolution, Figure 13C,D). μCT thus offers unique 3D insight into the microfacies of the sediment, albeit with limitations (Figure 13; [201]). Classical microfacies analysis is performed on thin sections, which are prepared following dehydration and impregnation of a sediment slab [202]. They allow a high-resolution study of the sedimentary components and how they relate to each other and remain the state-of-the-art method for microfacies analysis (Figure 13E). Recent developments with high-resolution hyperspectral imaging associated with machine learning algorithms also allow semiautomatic sedimentary structure discrimination in lake sediment cores [203,204].

Several event deposits are characterized by specific sedimentary structures, which can thus often be used to identify them. MTDs often contain strongly deformed sediments, including faults, folds, and mud-clast conglomerates (Figure 13D; [89,92,205,206]), and these structures are thus an indication of a mass wasting event. However, sometimes large internally undisturbed blocks can be included in an MTD and give a false impression in a sediment core of undisturbed in situ sediments [207]. Turbidites classically can contain ripples, convolute and parallel laminations, and erosive structures at their base (Figure 13A [24,26]). Lacustrine turbidites, however, are often composed of mud and merely show a graded to homogenous structure (Bouma Te division; [89]). However, in turbidites with a coarse silty to sandy basal part (Bouma Tc-Td), cross, convolute, and

parallel laminations have been recognized (e.g., in fjords [15]), although μCT imaging may be needed to visualize these structures (Figure 13A; [18,208]). As these sedimentary structures are a clear indication of deposition during a flow process, they can, for example, be used to distinguish a fallout tephra deposit (massive) from volcanic material deposited or reworked by a turbidity current (cross or parallel laminations) (Figure 13B). Further, 3D-microfacies analysis of such basal turbidite sands has also permitted the determination of the grain orientation and thereby the paleoflow direction of separate sand layers within a single (amalgamated) earthquake-induced turbidite [209] or for tsunami [210] and hurricane [211] deposits. Microfacies analysis of event deposits using thin sections and μCT scans has further enabled researchers to visualize bioturbation at the top and base of turbidites (Figure 13A; Figure 11A; [18,208]), to identify small-scale grading in turbidites [15,208], to identify internal mud clasts [126], to assess the state of fragile diatom frustules [15], and to detect (crypto)tephras (Figure 13A; [15,18,212]).

SSDSs are probably the event deposits for which sedimentary structures are the most crucial. They are in situ deformations and can thus not be distinguished from the background sediments based on their composition. SSDSs can exhibit a broad range of structures, from linear waves or disturbed laminations, over folds, faults, and liquefaction structures, to intraclast breccias (Figure 13C; [20,83,213]). The nature and grade of deformation are related to the sediment type, the thickness of the affected layer, and seismic ground motion and have been used to quantify the latter [20,83,214]. Avşar et al. (2016) developed a deformation index to objectively quantify the degree of deformation.

### 3.2.3. Inorganic Geochemistry

Most event deposits in lake sediments reflect the input of a detrital fraction that differs from the continuous (background) sedimentation. Inorganic geochemistry is thus a perfectly suited tool to detect and characterize event deposits, especially when it can be measured at high resolution. Indeed, inorganic geochemistry is known as a good descriptor of the composition of the sediment since the premises of modern limnogeology (e.g., [215]). However, the cost of such techniques, both in time and money, made it difficult to apply at a resolution high enough to detect event sedimentation. As a result, until the early 2000s, paleolimnologists mostly used magnetic mineralogy over inorganic geochemistry to support the visual description of event deposits. The development and success of geochemical XRF core loggers [216–218] radically changed the game by offering the possibility to both document and characterize any sediment layer that differs from the continuous sedimentation in a unique time-effective measurement run. The basic assumption here is that event deposits are characterized by a marked change in the sediment chemical composition when compared with the "continuous" sedimentation [30,46,200,219,220]. Even if they efficiently provide qualitative high-resolution profiles of elements composing the sediment, data obtained by XRF core scanning should be log-transformed, and possibly calibrated with traditional measurements of elemental concentrations, to be used quantitatively ([214]; Bertrand et al., this issue).

The reasons behind sediment inorganic geochemical changes in event deposits may be grouped into three different categories: (i) the composition and/or provenance of the sediment changes, (ii) grain size varies and is reflected in geochemistry through mineralogical sorting, and (iii) the event induced a change in the redox conditions that led to the precipitation of specific minerals (Figure 14).

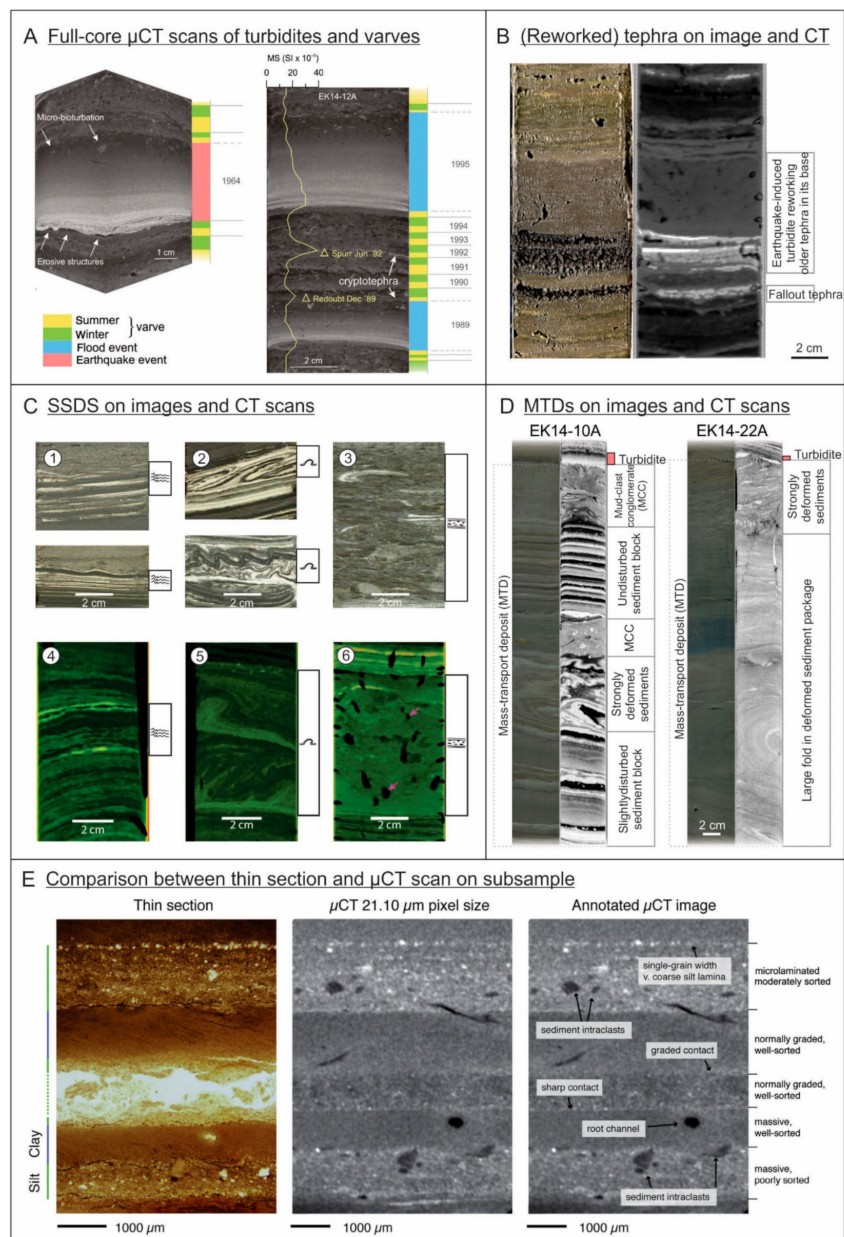

**Figure 13.** Examples of sedimentary structures and fabric in sediment cores. (**A**): µCT scan of a full core (35 µm voxel size) of Eklutna Lake (Alaska; modified after [18]) with, on the left, evidence of erosion by an earthquake-induced turbidite and microbioturbation in the top of the turbidite and in the varves. On the right are two flood-induced turbidites between bioturbated, varved sediments. Two cryptotephras that are identified using magnetic susceptibility can also be seen on the µCT scans. (**B**): Image and medical CT-scan frontal view of a core in Lake Calafquén (Chile; modified after [194]) with a fallout tephra that is reworked in an overlying turbidite. The CT data reveal the more random fabric in the fallout tephra compared to a laminated fabric of the tephra reworked by the turbidity current. (**C**): Images (1–3) of soft sediment deformation structures (SSDSs) in the Dead Sea (Israel; modified after [83,213]) and X-ray CT scans (4–6) of SSDS in lakes Riñihue and Calafquén (Chile; modified after [20]). Images 1 and 4 show examples of linear waves and disturbed lamination; 2 and 5 show folded layers; 3 and 6 show intraclast breccias. (**D**): Images and medical CT-scan frontal view of MTDs in Eklutna Lake (Alaska; modified after [18]). Both cores show the upper part of an MTD, including strongly folded and deformed sediments, undisturbed blacks, and mud-clast conglomerates (MCCs), covered by a turbidite. (**E**): Comparison between a thin section and µCT scan (21.10 µm voxel size) from glaciolacustrine varves from Glen Roy (Scotland; modified after [201]).

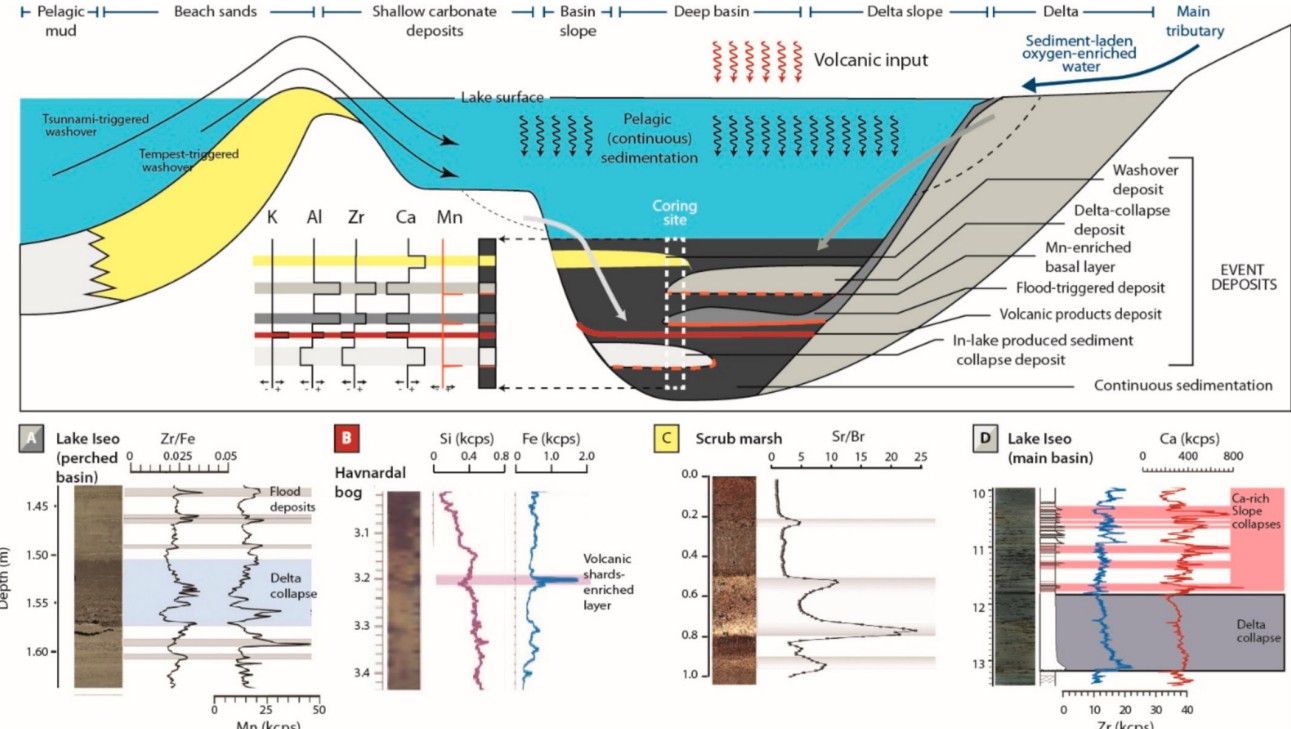

**Figure 14.** Schematic representation of event deposits that can generally be identified using inorganic geochemistry. The chemical elements represented here were chosen to represent a specific source or process: K is enriched in acidic volcanic deposits; Al is a typical marker of the siliciclastic fraction; Ca indicates the presence of carbonates, Zr usually tracks high-energy deposits due to the high density and grain size of zircon, and Mn is a good marker of reoxygenation events. The lower panel displays examples from the literature where downcore geochemical variations were used to: (**A**): Identify and distinguish flood deposits and delta collapses [29]; (**B**): Identify a thin volcanic deposit (here, through Fe enrichment; [75]); (**C**): Identify washover deposits in a coastal lake [16]; and (**D**): Discriminate between sources of mass wasting deposits [110].

Composition and/or Provenance of the Sediment Changes

Since sediment inorganic geochemistry is a bulk measurement, the concentrations of the various inorganic elements primarily reflect the composition of the sediment, which can be subdivided into the siliciclastic fraction, carbonates, organic matter, and biogenic silica (Bertrand et al., this issue). Most inorganic elements occur in the siliciclastic fraction, but carbonates and biogenic silica are important phases for Ca and Si, respectively. Organic matter, on the other hand, mostly decreases the concentration of inorganic elements through dilution.

Changes in the sediment composition constitute the most common cause of bulk sediment geochemical changes. In lake basins where sedimentation is dominated by the settling of authigenic particles (i.e., particles produced within the lake, mainly authigenic carbonates, biogenic silica, and organic matter), the input of allochthonous matter is characterized by a change in geochemistry that depends on the nature of the eroded material. In the case of a flood event, geochemistry reflects the geology of the siliciclastic eroded terrains (Figure 14A; [29]). In most cases, major (Al, K, Fe, Na), minor (Ti), or trace (e.g., Rb, Zr) elements associated with aluminosilicates generally allow the identification of flood deposits [46,220–223], especially in lakes with background sediments rich in organic matter, autochthonous carbonates, or biogenic silica. The erosion of carbonate terrains produces a small amount of fine sediment due to the high solubility of carbonates exposed to meteoric conditions.

Floods are not the only events supplying lakes with allochthonous sediment. Volcanic eruptions, for example, may also produce event deposits with distinct geochemical

signatures, whether they represent direct atmospheric fall-out or indirect input from lake tributaries (Figure 14B; [75]). In both cases, the volcanoclastic layers generally differ from the continuous sedimentation through an excess of elements that depend on the nature of the volcano. These elements are generally Ca, Fe, and Mg for products of volcanoes with a basic magma composition and Si, Al, and K in the case of acidic volcanism [75]. In the latter case, excess K is often the most reliable indicator since Si and Al tend to be ubiquitous in lake sediments.

In coastal lakes, high-energy events occurring in the adjacent sea may also supply considerable amounts of allochthonous sediment during marine washover, such as those caused by tsunami- and storm-derived giant waves [11,16,139]. Independently of the triggering mechanism, these events may be identified using inorganic geochemistry since they provide predominantly inorganic material from the near shore, beach, and/or dune system to generally organic-dominated ponds and lagoons. The geochemical signature of the associated deposit then depends on the composition of the dune system. They are typically enriched in Ca (and Sr) if the dune sands are dominated by shell fragments (Figure 14C; [219]) or in Si (and Zr) if the dune system is dominated by siliceous sands [155]. Other geochemical tsunami indicators include elements that are enriched in the marine environment, such as S, Cl, and Br [16,224].

In-lake reworking processes may also produce event deposits in lake basins. In general, only the steepest slopes are affected by such processes. The geochemistry of reworked sediments depends on the composition of the material that has been reworked. In the case of a delta collapse, the sediment composition is essentially the same as that of the tributary catchment area and flood deposits. In such a case, reworked layers can be identified using the same elements as for flood deposits [110]. In contrast, when nondeltaic lake slopes are reworked, the sediment geochemistry reflects the nature of the autochthonous sediments. For instance, the deposits associated with the reworking of lake carbonaceous platforms are enriched in Ca and thus difficult to distinguish from the Ca-dominated continuous sedimentation of carbonate-rich lakes. However, it has been shown that even in the case of high autochthonous calcite production, temperate lake basins do not accumulate carbonate below a certain depth [225]. Such Ca-enriched layers in the deepest part of the lake may be interpreted as resulting from the collapse of shallower Ca-rich lake slopes (Figure 14D; [110]).

Grain-Size Variability

The second way in which inorganic geochemistry can help identify event deposits is through its dependence on grain size via mineralogical sorting. Resistant minerals, such as quartz and zircon, tend to be enriched in the coarser fraction of the sediment [188]. This typically results in higher Si and Zr concentrations in the coarse fraction of the sediment. Clay minerals, on the other hand, tend to be enriched in K. In lake sediments, coarser deposits, therefore, tend to be enriched in Si and Zr, whereas higher K concentrations reflect abundant finer-grained particles. Since most event deposits have a typical grain-size signature (see Section 2), trends in elements related to grain size generally allow identification of event deposits. This is typically done using elemental ratios (or better log ratios) of a grain-size-related element (Si, Zr, Ti, Fe) versus a conservative element (typically Al or Rb). This approach avoids the influence of dilution by other components of the sediment and, therefore, best reflects the sediment grain size. However, such ratios tend to be site specific since they depend on the mineralogical composition of the sediment sources.

In three distinct lakes from the European Alps, for example, three distinct ratios were identified as grain-size indicators: Ca/Fe [46], Zr/Fe [19], and Fe/K [226]. Interestingly, this suite of ratios all involve iron (Fe), but two of them consider Fe as a marker of fine sediment, whereas one considers it representative of the coarsest particles. This illustrates the strong case dependency of such an approach for which there is no universal recipe. The use of elemental ratios as grain-size proxies must be used with caution and always after a detailed sedimentological, mineralogical, and geochemical study [188]. In addition,

inorganic elements are generally not linearly related to grain size over the entire grain-size range observed in event deposits. This requires the use of multiple elemental proxies, which better reflect the sediment grain size over a large size range ([185]; Bertrand et al., this issue). When applied to flood deposits, such an approach has not only allowed identification of flood turbidites but also proposed an interpretation of each flood layer in terms of the flood intensity [29]. The close relation between sediment inorganic geochemistry and grain size also allows the use of geochemistry to distinguish between the triggering mechanisms at the origin of event deposits. For instance, delta collapses generally rework the same material as floods, but their energy is generally higher than that of floods, resulting in a higher capacity to transport coarse particles. Therefore, the geochemical signature of delta collapses tends to be enriched in elements that reflect coarser (Si, Ca) or denser (Zr) minerals compared to flood deposits [110].

The Event Induced a Change in Redox Conditions

Finally, inorganic geochemistry may also track event sedimentation based on the precipitation of redox-sensitive elements. Under normal conditions, the biochemical functioning of lakes generally leads to an oxygen depletion of the water–sediment interface. Event deposits, on the other hand, are often accompanied by significant water-mass movements that are susceptible to quickly modifying the redox conditions at the sediment–water interface, leading to the precipitation of characteristic redox-sensitive chemical species (Figure 14A; [17,29]). Here, manganese (Mn) is particularly interesting because it is abundant in sediments and tends to dissolve under anoxic conditions and precipitate under oxic conditions. Mn occurs as Mn(IV) and diffuses and precipitates as carbonate Mn oxides at the oxic–anoxic boundary interface [227,228]. This process leads to Mn enrichment at the oxygen-rich water–sediment interface [227]. Under anoxic bottom waters, Mn(II) diffuses to the water column, and Mn becomes depleted in the sediments. Thus, the presence of Mn in the graded layers associated with terrigenous elements (Figure 14A) suggests oxygenated water input during turbidity currents, such as flooding, while reworked sediment linked to earthquakes does not necessarily present such enrichment [17]. Peaks in Mn have hence been used to identify the base of event deposits. However, if slope destabilization is triggered at a depth above the oxycline in stratified lakes, these mass movements could also bring oxygen to the lake bottom and induce a change in the redox conditions, as observed in Lake du Bourget [204] and Lake Iznik (Gastineau et al., under review).

3.2.4. Petrology

The petrology, i.e., composition, of event deposits can provide important information regarding their provenance and, in some cases, their mode of transport and deposition. The sediment composition can be analyzed either microscopically (binocular lens, petrographic microscope, SEM) or analytically (e.g., X-ray diffraction). Often, the simple observation of a smear slide on a petrographic microscope can already provide relevant information. Several books, e.g., [229], and online tools (e.g., [230]) can help to identify the most common components.

Petrology is particularly important for the identification of volcanic deposits. The presence of glass shards is the main criterion to attribute a volcanic origin to an event deposit (Figure 15A; [231]). However, the presence of volcanic glass does not automatically mean that the deposit is a tephra since background sediments in volcanic regions often contain dispersed volcanic particles, including glass shards. Consequently, different types of event deposits in volcanic regions may also contain glass shards.

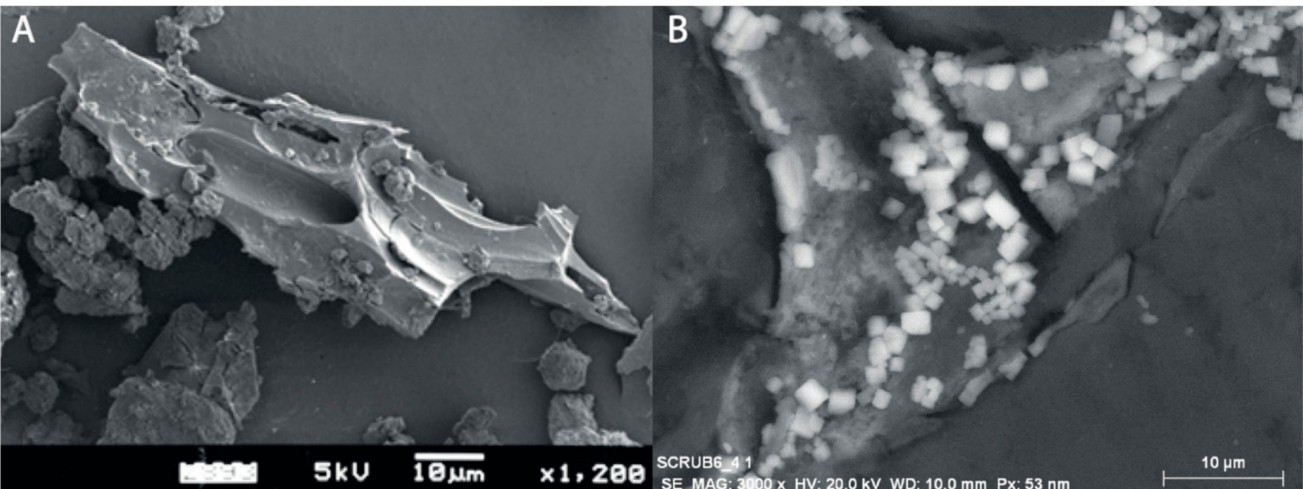

**Figure 15.** SEM images of particles characteristic of specific types of event deposits. (**A**): Glass shard, which demonstrates the volcanic nature of the deposit [231]; (**B**): Halite, also evidenced by XRD analysis, at the surface of two carbonated sand grains with a typical cubic shape. The presence of halite indicates a barrier origin of the sand, and thus a hurricane deposit, instead of a tsunami, which would contain particles reworked from a deeper environment [16].

In coastal lakes, the presence or absence of salt coatings on sand grains, indicating a barrier or lower-shoreface origin for sand grains, respectively, can also be used as a criterion to identify tsunamis (reworking both barrier and shoreface sediments) from hurricane (mostly reworking barrier sediments) deposits (Figure 15B; [16]).

### 3.3. Biological and Organic Facies

### 3.3.1. Organic Geochemistry

The amount and source of organic matter in lake sediments can serve as a first-order indicator of the presence of an event interrupting continuous sedimentation. Organic matter properties can also provide clues to identify the possible triggering mechanism. Often, the organic matter content is estimated using loss-on-ignition (LOI) at 550 °C [232]. For sediment with a low organic matter content (<2–3%), however, measuring the total organic carbon (TOC), which corresponds to ~50% of the total organic matter [233], with an elemental analyzer is more precise since LOI is also influenced by interstitial water in clays. Decarbonating the samples prior to TOC analysis is an essential pretreatment, e.g., [234]. Alternatively, TOC can also be estimated using the Compton/Rayleigh (inc/coh) scattering of the XRF core scanning dataset [235], visible-light reflectance spectroscopy [18,236], or hyperspectral imaging [237]. Another advantage of using an elemental analyzer is that it enables the simultaneous analysis of total nitrogen, hence the calculation of the C/N ratio, which permits distinguishing organic matter of aquatic (low C/N) and terrestrial (high C/N) origin, e.g., [238]. For quantitative estimates of the proportions of terrestrial and aquatic end-members, e.g., [239,240], using the N/C ratio instead of C/N [241] is recommended.

In addition to bulk organic elemental geochemistry, the stable isotopic composition of carbon ($\partial^{13}$C) allows distinguishing organic matter of marine ($\partial^{13}$C: −18 to −20‰) versus lacustrine ($\partial^{13}$C: −26 to −30‰) origin [238,242]. Although it is of limited use in strictly lacustrine environments, it is particularly suited to the identification of hurricane deposits [142,243]. Likewise, biomarkers, e.g., [244], and Rock-Eval pyrolysis, e.g., [245], can also help identify the nature of the deposited organic material and, therefore, distinguish between types of event deposits.

Flood turbidites are generally organic rich (e.g., [19,246]), except for those that represent glacial lake outbursts, which have very low TOC values (0.2–0.3%) due the glacial nature of the sediment (Figure 16A,B). Although flood and earthquake turbidites may have similar organic matter contents, flood turbidites typically have a purely terrestrial signature (C/N > 15–20), whereas the remobilization of hemipelagic slope sediments during earthquakes typically leads to a more aquatic signature (Figure 16A; [95]). This difference is particularly clear in lakes where aquatic productivity is significant [18]. C/N results must, however, be interpreted carefully since they are also affected by grain size, i.e., terrestrial plant remains are more abundant in the coarse fraction of sediments, and algal remains predominantly occur in the fine fraction, e.g., (Figure 16C; [247]). In addition, earthquakes may also trigger landslides and, therefore, remobilize terrestrial deposits, resulting in the deposition of earthquake-induced or postseismic (catchment response) turbidites with organic carbon predominantly of terrestrial origin [95]. Direct ash-fall deposits (tephra), on the other hand, tend to have a very low organic matter content (Figure 16D). In thick tephra deposits, most of the organic content represents posteruptive fine-grained lake sediments that percolated between coarser tephra grains, e.g., [71].

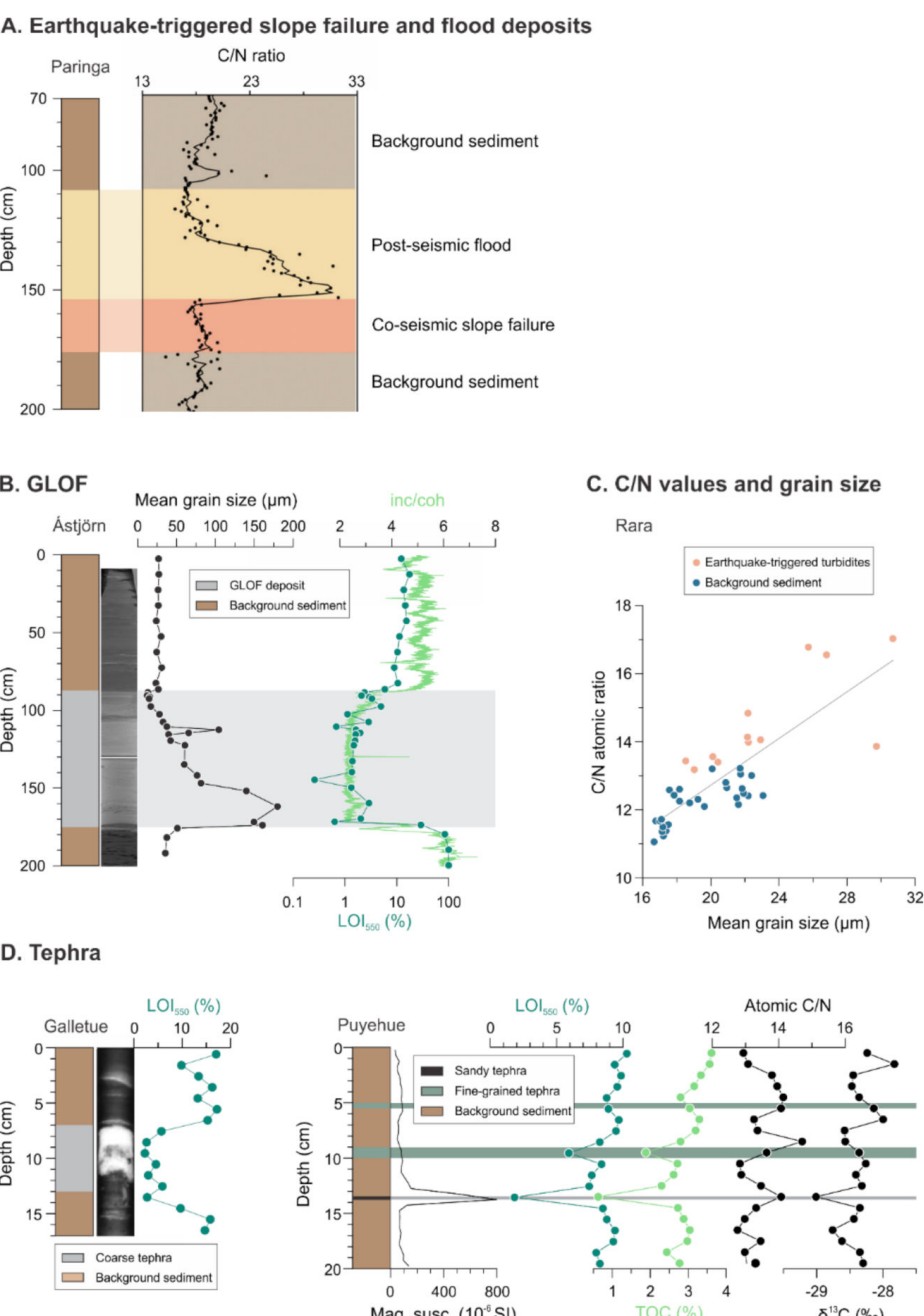

**Figure 16.** Examples of how bulk organic geochemistry can help identify event deposits in lake sediments. (**A**): Coseismic slope failure and postseismic flood deposits in the sediments of Lake Paringa, New Zealand [95]. The C/N ratio clearly shows that the flood turbidite contains more terrestrial organic carbon than the slope failure deposit, which has the same composition as the background sediments. (**B**): GLOF deposit in the sediments of Lake Ástjörn, Iceland [51]. The data illustrate the low-TOC nature of the glacial sediments and the potential of XRF inc/coh data to estimate the organic matter content. (**C**): Relation between the C/N ratio and sediment grain size for background sediments and earthquake-induced turbidites in Lake Rara, Nepal [247]. Coarser samples tend to be enriched in terrestrial organic matter (higher C/N) compared to fine-grained sediments, which typically contain more aquatic organic matter. Overall, the organic matter in the earthquake-induced turbidites is slightly more terrestrial than the background sediments, most likely because they represent the reworking of sediments deposited at shallower locations (closer to the lake shore). (**D**): Organic matter in and around tephra layers representing 20th-century eruptions of volcanoes in southern Chile: lakes Galletue [71] and Puyehue [233,248]. The latter example shows that tephras dilute lake organic matter but do not change its geochemical properties.

### 3.3.2. Biological Remains

Biological remains, including fauna and vegetation remains, are key criteria for identifying anomalous layers within a given sediment succession. Organic macroremains are generally abundant in lake sediments. They tend to be enriched in event deposits, especially within the coarse base of turbidites, where they represent material reworked from near-shore environments (see Section 3.3.1) (Figure 17B). Basically, their allochthonous origin in a given sedimentary environment demonstrates significant sediment transport and reworking. They are also of critical importance for radiocarbon dating, although reworked deposits should be avoided (i.e., [143,150,249]). Optical microscopy can be used to identify and quantify the origin of vegetation remains and can help in the identification of event deposits. Flood deposits are generally characterized by purely terrestrial signatures, whereas earthquake-related remobilization of hemipelagic slope sediments typically leads to a more lacustrine organic fraction [245]. The identification of vegetation debris, including clast of peat, blades of dune vegetation, and woody debris in a sand sheet deposited in a coastal lake of southeast Australia, is indicative of considerable vegetation stripping from the dune and back barrier and provides arguments supporting the tsunami hypothesis [250].

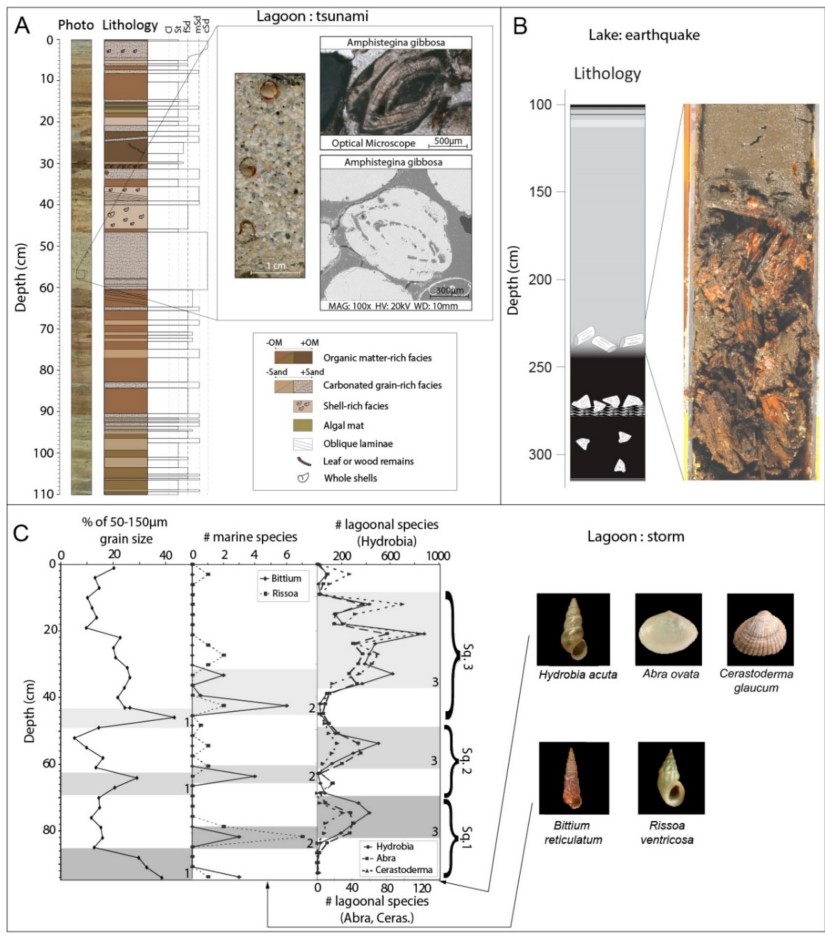

**Figure 17.** (**A**): Presence of reworked large abraded foraminifera (Amphistegina gibbosa) in a tsunami sandy deposit from the Scrub Island lagoonal system [16]. (**B**): Presence of coarse wood fragment at the base of an ~2-m-thick homogenite (Lake Icalma, Chile; [94]). (**C**): Sediment sequence of 3 sequences (Sq. 1, 2, 3) of storm deposits in Pierre Blanche lagoon identified as in increased grain size, then the appearance of marine species and finally lagoonal species [139]. Marine species are Bittium recticulatum (© National Museum Wales) and Rissoa ventricosa (© Desmarest–specimen from Le Brusc); lagoonal species are Hydrobia acuta (© Natural History Museum Rotterdam), Abra Ovata (© 2022-G&Ph-Poppe), and Cerastoderma glaucum (© Amgueddfa Cymru–National Museum Wales).

The fauna content in storm sediments consists of skeletal remains, shells [139], and/or shell fragments [136,150]. It is basically subdivided into macro- and microfauna remains. Within the macro-remains category, mollusks are particularly useful, as their shell remains are frequently well preserved. Moreover, they are large-sized and relatively easy to collect and identify. Up to 74% of known living bivalves are known as fossils [251,252]. Thus, their ecological requirements are accurately studied, and molluscan assemblages are frequently used to reconstruct coastal paleoenvironments [253]. They can be found in coastal lakes and they have been used to reconstruct coseismic subsidence [254]. Examples are given by the French Mediterranean lagoons, where storm events are recorded by marine mollusks (*Bittium reticulatum* and *Rissoa ventricosa*), in contrast with lagoonal ones (*Cerastoderma glaucum, Abra ovata*, and *Hydrobia acuta*) [139,149,151] (Figure 17C). Other examples are found in Lake Shelby (Alabama), where the presence of *R. cuneata* shells indicates that they were transported to their present location because these mollusks do not presently live in this coastal lake [142]. The preferred water depth of living mollusks can provide useful constraints on the origin of high-energy shell accumulations in or around coastal lakes. Offshore shells can be used as an argument supporting deep sea bottom reworking and a tsunami origin for high-energy deposits [255].

In coastal environments, foraminifera play a particularly important role in the fauna remains category. Most of the foraminifera are marine, and their presence in coastal lake sediments is a valuable indicator of marine flooding [16,140] (Figure 17A). Many living benthic foraminifera are stenobathes (organisms living in restricted bathymetric ranges), and their distribution along the shoreline is located at specific tidal elevations [256,257]. Thus, they can be used as indicators of the depth or distance of reworking by waves during storms or tsunamis. For example, marine foraminifera originating at least 5 km offshore have been found in modern and ancient storm deposits [140]. Abrasion of foraminifera tests can be linked to transport [258]. In sheltered settings where cohesive sediments are dominant, abrasion cannot be produced by wave action or by wind, and abrasion of benthic foraminiferal tests is likely the result of transport that can be an indicator of marine flooding [158]. In more exposed settings, such as sandy barriers between the open sea and coastal lakes, benthic foraminifera found in the lake can show higher diversity compared to sheltered settings. This can be explained by the fact that waves during storms may be large enough to erode and transport sediment and fauna from both shallow and deep environments [158].

Other microfossils, such as ostracods [259–261] or diatoms [262], are used for characterizing storm deposits preserved within the back-barrier lagoonal sediment. A detailed review of the use of microfossils to document past extreme tsunami and hurricane events is provided by [263]. Marine diatom species are also indicative of marine flooding of freshwater coastal lakes. Such polyhalous species were found in the Storegga tsunami deposits within a coastal lake of the Faroe Islands [264].

### 3.4. Other Approaches

Additional methods can be used on a case-by-case basis to identify event deposits and decipher their triggering mechanisms. This paragraph will not provide an exhaustive review of these methods, but it presents some useful selected case studies. A provenance approach based on mineralogy or radiogenic isotopes (Pb, Nd, Sr, Hf) can be applied to identify the origin of material composing event deposits and thus related transport mechanisms [265,266]. For instance, Sabatier et al. [155] used clay mineralogy in a coastal lagoon to identify storm deposits because the sandy barrier presents a contrasted clay mineral composition from the watershed, allowing the use of the clay mineral ratio to track marine submersion. A similar approach can also be applied for flood deposit identification [265,267]. Magnetic proxies such as anisotropy of magnetic susceptibility (AMS) provide valuable methods to identify increased foliation in a homogenite-type deposit compared to that of continuous sedimentation. Such foliation of the fine fraction is then related to specific flow conditions that have been linked to a seiche effect and thus probably

an earthquake event [30,219]. Molecular techniques such as sedimentary DNA are also promising to accurately discriminate between modern tsunami and storm deposits in coastal systems [268,269].

## 4. Chronology

The ability to accurately date event deposits in lake sediments is certainly as important as identifying them and deciphering their triggering mechanism(s). Chronology is also paramount to establishing the recurrence rates of certain types of events and the verification of the triggering interpretation via historical documentation. Sediment chronology is, therefore, one of the most critical aspects of event deposit research. Different types of chronological methods can be used to date recent lake sediments, such as radiocarbon, short-lived radionuclides, varve counting, palaeomagnetic secular variation, and historical events. The most widely used method to obtain sediment core chronologies is AMS radiocarbon dating of organic macro remains, including calibration with the most recent curve (IntCal20 [270] or SHCal20 [271]). However, organic macroremains need to be carefully selected depending on the lake settings. As a general rule, well-preserved leaves of terrestrial plants are better suited than wood fragments or pollen grains, which may have been stored in soils prior to reaching the lake [272]. In hardwater lakes, aquatic remains could present reservoir effects [273–275] and should, therefore, be avoided. Likewise, reworked materials, and thus sampling within event deposits [30], should be avoided, as they can merely provide maximum ages. The latter demonstrates the importance of conducting a detailed sedimentological investigation before selecting material for radiocarbon dating of lake sediments [276]. Radiocarbon ages obtained from carbonate samples such as shells are also used in coastal lakes, but a precise estimation of the reservoir age is required before age modeling [249]. Bulk samples, used when no macroremains are found, are generally older than the real age of deposition, even in regions free of carbonate and carbonaceous rocks. They generally represent inputs of preaged terrestrial organic carbon from soils in the lake watersheds. In some cases, the N/C ratio, pyrolysis organic geochemistry, organic petrography, and carbon stable isotopes of bulk organic matter can be used to correct bulk ages [277,278]. Alternatively, radiocarbon analysis can also be performed on organic compounds (biomarkers) derived from plants that utilize atmospheric $CO_2$ [279–281] or even aquatic $CO_2$ [282].

In sediment cores containing event deposits, the first step before age modeling is to identify all event deposits with a reasonable thickness threshold (typically 1 or 5 mm) to build an event-free sediment depth [40,182,196]. These event deposits interpreted as instantaneous events need to be excised to avoid time evolution in the event deposit itself when modeling is applied. The second step is to run an age model on this event-free depth, with all ages corrected from the cumulative event deposit thickness above their respective depths. The final step is to reintegrate all event deposits at their respective depths in the best age model and to attribute the same age to the whole thickness of a given event deposit. These last two steps can either be carried out manually or by indicating the upper and lower limits of all event deposits in age–depth modeling software such as *clam* [283] and *Bacon* [284]. The latter method, however, tends to be restricted to sediment sequences with relatively low numbers of event deposits. In either case, each event deposit is assigned a single age, which is represented on the final age model with vertical bars (instantaneous deposition), together with uncertainties (2s) derived from the age–depth modeling simulations that incorporate the uncertainty of, e.g., $^{14}C$ ages (Figure 18A).

For example, the age–depth model of Lake Savine (Figure 18A) was constructed on the continuously deposited sediments (200 cm in total) of a 6-m-long core after removing 220 event deposits with a thickness of 5 mm or more, representing a total of 391 cm [17]. Note that SSDSs, such as in situ folded layers, intraclast breccia layers, and microfaults, should not be removed before age modeling [83], as the thickness of the originally affected sequence is unclear. Moreover, event deposits that relate to energetic flow processes (e.g., megaturbidite and MTD) could lead to in-lake erosion of previously deposited sediment, and thus, a potential hiatus has to be carefully evaluated during age modeling [30].

Short-lived radionuclides based on the measurement of natural $^{210}$Pb and $^{226}$Ra and artificial $^{137}$Cs and $^{241}$Am activities provide the most accurate and widely used chronological technique for the past century. Several models can be used to build a chronology from these data [285]. The first objective is to produce a chronology based on the excess $^{210}$Pb activities ($^{210}$Pbex) and then compare this chronology with the ages of artificial peaks in $^{137}$Cs and $^{241}$Am, which are byproducts from nuclear weapons tests conducted from 1955 to 1963 and by the Chernobyl accident in 1986 for the Northern Hemisphere or other nuclear tests and accidents [286]. Here, properly identifying event deposits is crucial to generating accurate chronologies for lake sediments. Event deposits must be removed before age modeling because they classically present lower $^{210}$Pbex activities in relation to high input of old sediment (from the watershed or previously deposited in the lake) diluting the $^{210}$Pbex from the atmosphere; thus, they must be considered instantaneous [40]. These short-lived radionuclides measurements are particularly important for event-deposit studies because they allow the precise dating of recent event deposits, which can then be linked with historically documented events around the lake. This allows both support of the chronology and allow the identification of the sedimentary signature of known events. To build an age model from these data, the recent R package *serac* was developed and easily allows reproducibility of the main hypotheses behind any age–depth model, such as changes in sedimentation rates or the presence of event deposits (Figure 18B).

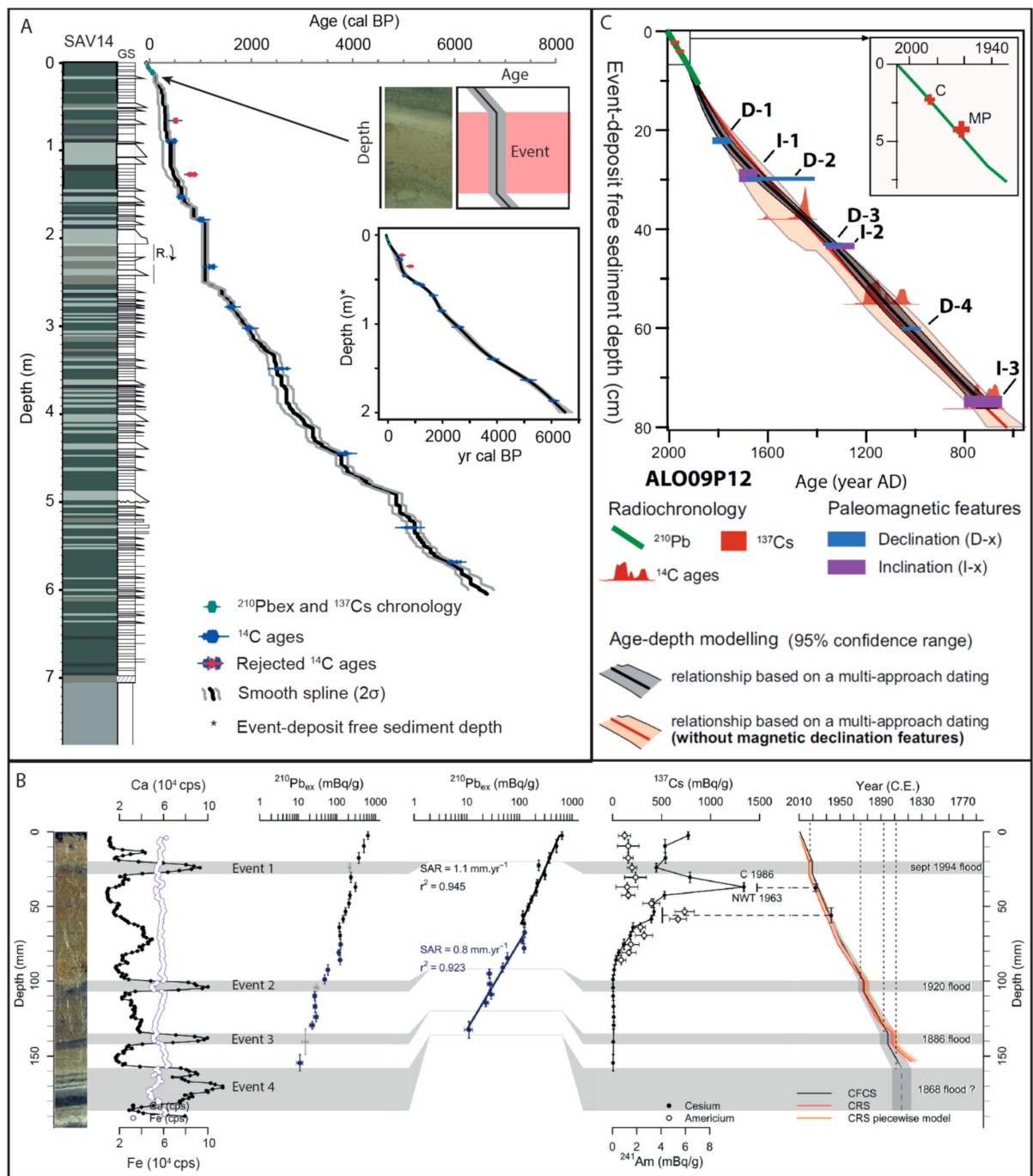

**Figure 18.** (**A**): Age-depth model of the SAV14 sediment core after removal of the 220 event deposits. The core chronology is based on radiocarbon dating and is short-lived radionuclides derived and was produced with the clam R package (**top right**). The final age–depth model (with event deposits) for the entire SAV14 sediment core is presented along the core lithology. The inset illustrates the shape of an age–depth model across an event deposit (constant age for the entire event thickness) [17]; (**B**): From left to right: core photograph, Ca and Fe content, $^{210}$Pbex activity with and without instantaneous deposits, $^{137}$Cs activity and $^{241}$Am activity, and the different types of $^{210}$Pbex age–depth models (CFCS, CRS, and CRS_pw) for the Lake Allos sequence [287]. The horizontal gray lines indicate layers that were identified as instantaneous deposits. (**C**): Age-depth model for the ALO09P12 sediment core, combining 137Cs activity peaks, sedimentation rates resulting from the 210Pbex-CFCS model, 14C ages, and magnetic features plotted against the event-free depth [288]. The age–depth model was produced with the clam R-code package, using the smooth spline function [283]. From five alpine lakes, including magnetic parameters in the core chronologies reduces age uncertainties by up to 30% [288].

A third group of analyses that can provide very accurate chronological data for lake sediments is varve counting, which is particularly important for high-resolution studies because it provides a clear and simple means to identify one year of deposition, e.g., [289]. This method is, however, restricted to lakes with strong seasonal contrasts in climatic and depositional conditions, resulting in seasonally distinct clastic, biogenic, or endogenic sedimentation [290]. In addition to establishing a precise age model [15,52,64,291,292], varves also allow investigation of the seasonality of event deposits such as floods [293]. For more details, a special issue about varved sediments was recently published in *Quaternary* [294].

Finally, other types of analyses can be used to refine age models, such as anthropogenic perturbations of known age, previously dated tephra deposits (tephrochronology), or the identification of historical events. Geomagnetic field secular variations are a complementary tool to establish more robust age–depth relationships, especially when the age model accuracy suffers from large radiocarbon uncertainties [288,295]. Once the quality of palaeomagnetic secular variations is verified, the declination/inclination variations observed in the sedimentary sequence are then correlated with the reference curve to provide additional stratigraphic chronomarkers (e.g., Figure 18C).

Combining results from different chronological techniques requires the use of age–depth modeling software such as *Bacon, Bchron, Clam*, and *OxCal*. The *Clam* R package employs classical statistical methods [283], whereas the three others use Bayesian statistics (Bacon [284]; Bchron [296]; OxCal [297]). These programs are able to integrate different types of age markers (short-lived radionuclide, tephra, historical event, etc.) in addition to radiocarbon ages. Wright et al. [298] compared the relative performance of these different programs for age modeling and found that no single modeling package outperforms the others, but a combined approach can exploit the strengths of each one of them to produce a 'consensus', illustrating that choosing the 'best model' is not a simple task [299]. Uncertainty estimations differ considerably among models but tend to be more realistic in models using Bayesian statistics [300]. A recent study suggests that Bayesian age–depth models become more precise than classic age models after a minimum dating density is reached and recommends the use of Bayesian age–depth models for a minimum of two dates per millennium [301].

In practice, we recommend combining different chronological techniques, such as $^{14}$C, $^{210}$Pbex, $^{137}$Cs, geomagnetic field secular variations, and identification of known tephra deposits and historical events, to produce the most realistic chronology with the lowest uncertainties. This is particularly true for sedimentary sequences with numerous event deposits, in which the potential for reworked deposits and organic material is particularly high.

## 5. Conclusions and Perspectives

Depending on the geological context, lake systems may contain event deposits related to one or more of the nine main processes discussed above (floods, glacial lake outburst floods, avalanches, volcanic eruptions, earthquakes, delta collapses, marine or in-lake tsunamis, cyclones, and hurricanes). Identifying the trigger of each event deposit is not always straightforward as different processes can lead to deposits with similar sedimentological characteristics (e.g., turbidites). Consequently, obtaining information on both the sediment origin and depositional processes is generally needed. This requires the use of a multi-method approach at the lake basin scale (geophysical survey, multiple coring), combined with a multiproxy analytical approach (sedimentology, geochemistry, geophysics, biotic approach) at the sediment core scale.

In this review paper, we summarized the different processes capable of inducing event deposits and we illustrated the most typical sedimentary facies of each type of event. We then described the most indicative proxies to determine the triggering mechanism(s) at the origin of event deposits in lake sediments. In addition, we emphasized the need to properly consider the instantaneous deposition of event deposits to obtain reliable chronologies on sediment cores containing event deposits. We recommend combining several chronological

techniques, which allow the generation of the most accurate core (and event) chronologies. Obtaining a precise chronology for the last decades is particularly important to compare the sediment records to instrumental data and/or historical observations, and hence obtain a better understanding of the sedimentary signature of event deposits preserved in sedimentary archives.

Perspectives for the future evolution of the field include the development of new high-resolution methods such as CT scanning, providing sedimentological fabric characterization [209], hyperspectral imaging providing automatic determination of event deposits at very high resolution [203,204] (see also Jacq et al., in this issue), and clustering and end-member modeling techniques based on grain-size or XRF core scanner data [302]. Investigating multiple lake systems in the same area is also encouraged to better estimate the spatial imprint of specific events (e.g., earthquake, flood, tsunami, etc.) [132]. Since lakes have distinct sensitivities to events, this may also result in a better definition of the location and/or magnitude of specific events [87]. Another approach is to integrate data from lake systems with other types of archives, such as tree rings, stalagmites, river deposits, marine sediments, trenches, or geomophology [169,303,304], with a special emphasis on nearshore environments [133,305]. Finally, a particular effort must be made to assess the influence of an event on the record of other types of events, such as the influence of tephra fall deposition on the record of turbidites [94] and the impact of large earthquakes on the record of subsequent higher-frequency/magnitude floods or debris flows [91,97,248]. A better understanding of this event interdependency is crucial to improve hazard assessments.

**Author Contributions:** All co-authors contributed to the conceptualization and writing of this review paper. All authors have read and agreed to the published version of the manuscript.

**Funding:** This research received no external funding.

**Institutional Review Board Statement:** Not applicable.

**Informed Consent Statement:** Not applicable.

**Data Availability Statement:** Not applicable.

**Acknowledgments:** We warmly thank Thierry Guyot for drawing Figure 1. Willem van der Bilt, Jamie Howart, Zakaria Ghazoui, Roberto Urrutia, and Alberto Araneda are acknowledged for providing some of the data used to make Figure 15. The authors thank doctoral students, postdocs and technical staff in our different research groups without whom this review paper would not be the same. Moreover, we thank all authors that perform high-quality studies on lacustrine event deposits.

**Conflicts of Interest:** The authors declare no conflict of interest.

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
