# Peer review of "A Review of Event Deposits in Lake Sediments"

_quaternary, doi:10.3390/quat5030034_

Round 1
Reviewer 1 Report
The review by Pierre Sabatier and colleagues of event deposits in lake sediments is a timely synthesis work of the abundant literature in the field. It will help the scientific community to reflect on the main problems, the unsolved questions and the new techniques and protocols to move forward. The review is comprehensive and the literature review is up to date. It is not an easy task to synthesize the topic but the authors have done a good job. Figures are illustrative and help to compare the different types of processes involved and triggering mechanisms. One minor detailed that could be change is the classification of event deposit type and mix the mechanisms behind at different scales. Earthquake and tsunami- related could be in the same category. And Delta Collapse could be caused by internal depositional processes or any other forcing mechanism. This paper is an opportunity to produce a clear classification of forcing mechanisms for event deposits in lakes and I would encourage the authors to change slightly Table 1.
The authors describe in detail how to identify event deposits and keys to be able to ascribe them to an specific mechanism. This section is the "core" of the manuscript and it will be quite useful to scientists in the same discipline. The paper includes many examples of event deposits in diverse contexts and illustrates the difficulties of clearly identifying them and reconstructing the depositional processes and the forcings.
Figures and pictures are of good quality. Hopefully they will be printed at large scale, so the fine details will not be missed. Reference list is exhaustive. The structure of the manuscript is correct. The chronological section is an honest exploration of the difficulties of obtaining accurate age models for lake sequences including event deposits.
Overall, the manuscript is a good synthesis of the state of the art in paleo-event deposits in lakes and it will be helpful to move forward for the community working on hazards and related topics.
Author Response
The review by Pierre Sabatier and colleagues of event deposits in lake sediments is a timely synthesis work of the abundant literature in the field. It will help the scientific community to reflect on the main problems, the unsolved questions and the new techniques and protocols to move forward. The review is comprehensive and the literature review is up to date. It is not an easy task to synthesize the topic but the authors have done a good job. Figures are illustrative and help to compare the different types of processes involved and triggering mechanisms.
Thanks for this nice comment
One minor detailed that could be change is the classification of event deposit type and mix the mechanisms behind at different scales. Earthquake and tsunami- related could be in the same category. And Delta Collapse could be caused by internal depositional processes or any other forcing mechanism. This paper is an opportunity to produce a clear classification of forcing mechanisms for event deposits in lakes and I would encourage the authors to change slightly Table 1.
The forcing mechanisms are described in the main text. The table 1 focuses on identifying the event deposits not to classification.
In the previous section about delta collapse we already specify: "The causes of these delta collapses are diverse and range from terrestrial mass movements impacting the delta plain [96] to floods [17], earthquakes [94,101] and sediment overload ("spontaneous”) [95,98]." Thus forcing mechanism is mostly linked to external processes and sometime internal.
For earthquake and tsunami- related events, even if there are mostly related to a seismic event, there are quite different phenomena and the event deposits have very different characters. For earthquake, it is mainly the seismic shaking itself (in-situ deformations) or triggering of gravitational mass movements (MTDs, turbidites…) in the deeper part of the lake, while sediment transport and deposition by tsunami is very different and mostly related to wave submersion and backwash processes and induced coastal deposits.
Finally, all co-author discuss and we do not want to change the current classification.
The authors describe in detail how to identify event deposits and keys to be able to ascribe them to an specific mechanism. This section is the "core" of the manuscript and it will be quite useful to scientists in the same discipline. The paper includes many examples of event deposits in diverse contexts and illustrates the difficulties of clearly identifying them and reconstructing the depositional processes and the forcings. Figures and pictures are of good quality. Hopefully they will be printed at large scale, so the fine details will not be missed. Reference list is exhaustive. The structure of the manuscript is correct. The chronological section is an honest exploration of the difficulties of obtaining accurate age models for lake sequences including event deposits. Overall, the manuscript is a good synthesis of the state of the art in paleo-event deposits in lakes and it will be helpful to move forward for the community working on hazards and related topics.
Thanks a lot for these nice comments
Reviewer 2 Report
Dear Sir! I have reviewed the article "A review of event deposits
in lake sediments" Pierre Sabatier1, Jasper Moernaut2, Sebastien
Bertrand3, Maarten Van Daele3, Katrina Kremer4, 2 Eric Chaumillon5,
Fabien Arnaud1. I want to say that this is a well-done work that
will be especially useful for students of geological specialties
and scientists involved in the study of modern sediments.
Undoubtedly, it should be published in Your journal. I found almost
no errors (with the exception of couple - line 224 rsulting -
resulting, 440 double quotes) and the absence of some spaces in the
text. I do not have any connections with the authors and I am
absolutely not bound by any obligations with them. Sincerely Yours
Author Response
Dear Sir! I have reviewed the article "A review of event deposits in lake sediments" Pierre Sabatier1, Jasper Moernaut2, Sebastien Bertrand3, Maarten Van Daele3, Katrina Kremer4, 2 Eric Chaumillon5, Fabien Arnaud1. I want to say that this is a well-done work that will be especially useful for students of geological specialties and scientists involved in the study of modern sediments. Undoubtedly, it should be published in Your journal. I found almost no errors (with the exception of couple - line 224 rsulting - resulting, 440 double quotes) and the absence of some spaces in the text. I do not have any connections with the authors and I am absolutely not bound by any obligations with them. Sincerely Yours
Many thanks to reviewer 2 for this nice comment, we correct these two wording mistakes
Reviewer 3 Report
It is an extremely interesting and useful review for anyone working in the area, with 305 references updated up to 2022. All important techniques used in the study of sedimentary archives for reconstruction of climatic events are mentioned and evaluated. I only recommend checking the legends of Figures 1, 2 and 10, which are incomplete.
Author Response
It is an extremely interesting and useful review for anyone working in the area, with 305 references updated up to 2022. All important techniques used in the study of sedimentary archives for reconstruction of climatic events are mentioned and evaluated. I only recommend checking the legends of Figures 1, 2 and 10, which are incomplete
Thanks to reviewer 3 for this nice comment, we check and complete the 3 legends, it is related to figure size.
Reviewer 4 Report
This review paper by Sabatier et al. summarizes and illustrates the sedimentary facies of 8 main processes that may trigger event deposits in lake sediments. Authors also describe the most indicative proxies of triggering mechanisms and outline practices in chronology for the lake sedimentary sequence study. Overall, this MS is well-written, presents a comprehensive summary of major processes and corresponding methodologies, and includes well-drawn figures that effectively demonstrate each depositional events. The paper subject is undoubtedly suitable for Quaternary, and I recommend the following revisions to the MS:
P5 L146-148: this distal underflow deposit is also shown in Fig 2, but the description of its sedimentary facies is missing in the manuscript. Are they homogenous silty or clayey? Or graded turbidites?
P7 L224: “rsulting” typo
P10 L310 section 2.5 Earthquakes: are the earthquakes discussed here all occur onshore or also include subaqueous earthquakes? Are there differences in sedimentary facies among the differently located earthquakes?
P10 L317: is there a summarized rough range of the magnitude threshold?
P15 L471: why is there no description of the sedimentary facies of backwash deposit? It is shown in Figure 8 though.
P21 L621: Figure 10C should be 10D
P21 L624: Figure 10D should be 10E
P21 L632: Figure 10F
P32 L953-956: they can be further differentiated by using methods described in L991-995, is it right?
P34 L1033: “online tools (e.g.,)” examples are missing
P35 L1036: “low amounts of volcanic glass” should be high amounts of…
P36 L1075: “Figure 15A,B” should be Figure 16
P38 L1113: “Fig 14” should be Figure 17
P41 L1193: “parmount” typo
P42 L1231: “Fig 16a” should be Figure 18A
P42 L1252: “than be” typo
P42 L1256: “Fig 15B” should be Figure 18B
P44 L1288: “Fig 15C” should be Figure 18C
Figure 1: first part of the figure caption is missing.
Figure 2: first part of the figure caption is missing.
Figure 6: explain the abbreviations of s.l. and s.s. in figure caption.
Figure 10: first part of the figure caption is missing.
Figure 12: to confirm, there is only one data point for avalanche deposit in Fig A, right?.
Figure 15: in the caption, the presence of halite should indicate tsunamis rather than hurricane.
Author Response
This review paper by Sabatier et al. summarizes and illustrates the sedimentary facies of 8 main processes that may trigger event deposits in lake sediments. Authors also describe the most indicative proxies of triggering mechanisms and outline practices in chronology for the lake sedimentary sequence study. Overall, this MS is well-written, presents a comprehensive summary of major processes and corresponding methodologies, and includes well-drawn figures that effectively demonstrate each depositional events. The paper subject is undoubtedly suitable for Quaternary, and I recommend the following revisions to the MS:
Many thanks for this very positive review
P5 L146-148: this distal underflow deposit is also shown in Fig 2, but the description of its sedimentary facies is missing in the manuscript. Are they homogenous silty or clayey? Or graded turbidites?
Thanks, we add in Figure 2 we add the sedimentary description as in the text : “as a relatively fine-grained turbidite”
P7 L224: “rsulting” typo
Corrected
P10 L310 section 2.5 Earthquakes: are the earthquakes discussed here all occur onshore or also include subaqueous earthquakes? Are there differences in sedimentary facies among the differently located earthquakes?
In the first sentence of this section we precise “onshore or subaqueous earthquake”. For earthquake localized onshore or in lake. The most important parameter is the seismic intensity. A study identifies different sedimentary imprint for different kind of earthquake such as megathrust or intraplate (Praet et al., 2020). In this section we add the following sentence: “By tracking the sedimentary imprint of well-documented historical earthquakes throughout multiple basins, it is inferred that the type and size of the sedimentary imprint is largely controlled by the local seismic intensity, i.e. the strength of seismic ground motion at the lake (Moernaut et al., 2014; Van Daele et al., 2015; Daxer et al., 2022). This implies that an earthquake source below the lake may produce similar imprints than onshore earthquake sources (Gastieneau et al., 2021) Moreover, the type and the size of sedimentary imprint can also be influenced by the duration and frequency content of seismic ground motion (Praet et al., 2022; Van Daele et al. 2019). In some cases, megathrust earthquakes (long duration and low frequency) facilitate the triggering of multiple, voluminous landslides and the generation of megaturbidites, while intraplate earthquakes (short duration and high frequency) may mostly induce onshore landslides, surficial slope remobilization and the generation of thinner turbidites.”
P10 L317: is there a summarized rough range of the magnitude threshold?
There is no magnitude threshold, because it depends on the lake sensitivity and we detail this in the following sentence of the manuscript.
P15 L471: why is there no description of the sedimentary facies of backwash deposit? It is shown in Figure 8 though.
Yes, there is a description of backwash deposit in the first version of the manuscript from L425-433. But few Example is available from bibliography.
P21 L621: Figure 10C should be 10D
Corrected
P21 L624: Figure 10D should be 10E
Corrected
P21 L632: Figure 10F
We add the reference to this figure
P32 L953-956: they can be further differentiated by using methods described in L991-995, is it right?
Yes, this is why we mention both methods.
P34 L1033: “online tools (e.g.,)” examples are missing
We add the reference to Myrbo, A., Morrison, A., and McEwan, R. (2011). Tool for Microscopic Identification (TMI). http://tmi.laccore.umn.edu. Accessed on 10 May 2022
P35 L1036: “low amounts of volcanic glass” should be high amounts of…
We delete “low amount”
P36 L1075: “Figure 15A,B” should be Figure 16
Thank, it is corrected
P38 L1113: “Fig 14” should be Figure 17
corrected
P41 L1193: “parmount” typo
Corrected
P42 L1231: “Fig 16a” should be Figure 18A
Corrected
P42 L1252: “than be” typo
Corrected
P42 L1256: “Fig 15B” should be Figure 18B
Corrected
P44 L1288: “Fig 15C” should be Figure 18C
Corrected
Figure 1: first part of the figure caption is missing.
corrected
Figure 2: first part of the figure caption is missing.
corrected
Figure 6: explain the abbreviations of s.l. and s.s. in figure caption.
s.l. : sensu lato (in the broad sense) and s.s. : sensu stricto (in the narrow sense), we add the definition in the figure caption
Figure 10: first part of the figure caption is missing.
Corrected
Figure 12: to confirm, there is only one data point for avalanche deposit in Fig A, right?.
No it depend of the size of the deposit.
Figure 15: in the caption, the presence of halite should indicate tsunamis rather than hurricane.
No this salt coating mostly indicate hurricane deposit we add the following sentence in the caption “…indicates a barrier origin of the sand and thus a hurricane deposit and not a tsunami which mostly present deeper reworked sediment”